# LEARNING FROM CONFLICTING DATA WITH HIDDEN CONTEXTS

## ABSTRACT

Classical supervised learning assumes a stable relation between inputs and outputs. However, this assumption is often invalid in real-world scenarios where the input-output relation in the data depends on some hidden contexts. We formulate a more general setting where the training data is sampled from multiple unobservable domains, while different domains may possess semantically distinct input-output maps. Training data exhibits inherent conflict in this setting, rendering vanilla empirical risk minimization problematic. We propose to tackle this problem by introducing an allocation function that learns to allocate conflicting data to different prediction models, resulting in an algorithm that we term LEAF. We draw an intriguing connection between our approach and a variant of the Expectation-Maximization algorithm. We provide theoretical justifications for LEAF on its identifiability, learnability, and generalization error. Empirical results demonstrate the efficacy and potential applications of LEAF in a range of regression and classification tasks on both synthetic data and real-world datasets.

## 1 INTRODUCTION

Classical supervised learning assumes a stable relation between inputs and outputs (Vapnik, 1999). On the other hand, real-world objects often have a variety of semantic properties, and which one is of interest depends on the context. Such contexts are implicitly embedded in the datasets during the labeling process in conventional supervised learning. However, contexts may not be accessible in many open-ended real-world scenarios due to their innate unobservability (Harries et al., 1998) or privacy and security constraints (McMahan et al., 2017; Mitchell et al., 2021). With hidden contexts, the same input may correspond to distinct outputs due to the change of the context, violating the stable relation assumption. For example, when collecting uncurated image-label pairs from the Internet for a general recognition task, an image of a red sphere may be sometimes labeled as "red" and sometimes "sphere". Similar cases are also identified across datasets (Taori et al., 2020): an image of a scuba diver can be labeled as "scuba diver" in ImageNet (Deng et al., 2009) while being labeled as "person" in Youtube-BB (Real et al., 2017). Contexts can also reflect human preferences whose identities are not available in privacy-sensitive applications, whilst influencing predictions (Kairouz et al., 2021).

It is easy to see that when the stable relation assumption is invalid, the common practice of performing global Empirical Risk Minimization (ERM) is problematic since the data exhibit inherent *conflict*: the outputs conditioned on the same input can be semantically distinct across different examples, thus mutually interfering during training. Existing machine learning models usually rely on human efforts to filter or re-label the data to eliminate this conflict (Krause et al., 2016; Vo et al., 2017; Taori et al., 2020). These workarounds typically require domain knowledge on the specific problem, and could be ineffective when some crucial side-information of the data is hard to define or collect in practice (Hanna et al., 2020). It is then natural to ask: can we empower the learner to *automatically* eliminate the conflict in the training data without additional human intervention?

In this work, we formulate the problem of training on conflicting data by assuming that the data comes from multiple unobservable *domains*; different domains may have semantically distinct input-output maps, while the map in each domain is stable. In particular, instead of learning a global model, our goal is to learn *multiple local models* that separately capture mutually conflicting input-output relations in data. To this end, we propose LEAF (LEarning from conflicting data by Allocation Function), a learning framework that comprises a high-level *allocation function* and a low-level model set. The allocation function learns to allocate the data among the models to eliminate the conflict,

while the models learn local predictions. Our key insight is that the conflict *itself* can be exploited to learn a good allocation strategy: if the allocation function assigns conflicting examples to the same model, then this will hinder the global minimization of training error due to the conflict, which can in turn be used to improve the allocations. Using a probablistic reinterpretation of our framework, we establish a connection between LEAF and a variant of Expectation-Maximization (EM) (Dempster et al., 1977), showing that an analytic form of the allocation function can be explicitly derived.

We provide theoretical justifications of LEAF by analyzing its identifiability, learnability, and generalization error. Our theoretical results generalize the analysis on classical single-domain supervised learning with no conflict (Vapnik, 1999), revealing the conditions under which LEAF can provably recover all conflicting concepts and providing a generalization error upper bound. Empirically, we conduct extensive experiments that span regression and classification tasks on both synthetic data and real-world datasets. Experimental results show that LEAF effectively resolves the conflict in the training data, significantly outperforms task-specific approaches in three settings reflecting practical scenarios with different label space structures, and achieves competitive results with *fully* supervised learning methods with domain index or label set oracles. In summary, our contributions are three-fold:

- We formulate a supervised learning setting of learning from conflicting data, which captures the key difficulty in training on the data with multiple hidden contexts.

- We introduce a theoretically grounded learning framework termed LEAF that automatically eliminates the conflict in data and establish its connection with a variant of the EM algorithm.

- We empirically evaluate LEAF on a wide range of tasks, in which LEAF effectively resolves the conflict in data without additional human intervention, outperforms task-specific approaches, and achieves competitive results even with fully supervised learning methods.

## 2 PROBLEM FORMULATION AND THE LEAF FRAMEWORK

In this section, we formulate our learning problem and introduce the LEAF framework. We adhere to the conventional supervised learning terminology: let $\mathcal{X} \subseteq \mathbb{R}^L$ be an input space, $\mathcal{Y}$ an output space, $\mathcal{H}$ a hypothesis space where each hypothesis (model) is a function from $\mathcal{X}$ to $\mathcal{Y}$, and $\ell : \mathcal{Y} \times \mathcal{Y} \to [0, 1]$ a non-negative and bounded loss function. We denote by $\lambda(\mathcal{S})$ the Lebesgue measure of $\mathcal{S}$ for any $\mathcal{S} \subseteq \mathcal{X}$. We use $[k] = \{1, 2, \cdots, k\}$ for positive integers $k$ and use $\mathbb{1}$ as the indicator function. For a probability density $P_X$ over $\mathcal{X}$, we denote by $\mathrm{supp}(P_X) := \{x \in \mathcal{X} \mid P_X(x) > 0\}$ its support. For a set $S$, we use $|S|$ to denote its cardinality. We use superscripts to denote sampling indices (e.g., $\mathbf{d}^i$ and $\mathbf{x}^{ij}$) and subscripts to denote element indices in a given ordered set (e.g., $d_i$).

### 2.1 PROBLEM FORMULATION

We begin by formalizing the notion of conflict. Following the seminal works in domain adaptation (Ben-David et al., 2010a; Mansour et al., 2009; 2008), we define a domain $d$ as a pair $\langle P_X, c \rangle$ consisting of an input distribution $P_X$[1] over $\mathcal{X}$ and a target function $c : \mathcal{X} \to \mathcal{Y}$. We assume that the training data is sampled from a *domain set* $D = \{d_i\}_{i=1}^N = \{\langle P_i, c_i \rangle\}_{i=1}^N$ containing $N$ (agnostic to the learner) domains, and *the goal of the learner is to learn **all** target functions* $\{c_i\}_{i=1}^N$. We focus on the setting where there exists conflict between domains, which we make formal by Definition 1.

**Definition 1** (Conflicting domains). *For two domains $\langle P, c \rangle$ and $\langle P', c' \rangle$, let $\mathcal{X}_{\mathrm{int}} = \mathrm{supp}(P) \cap \mathrm{supp}(P')$ be the intersection of the supports of their input distributions. We say that the two domains are* conflicting *if there exists $\mathcal{S} \subseteq \mathcal{X}_{\mathrm{int}}$ such that $\lambda(\mathcal{S}) > 0$ and $c(x) \neq c'(x)$ for every $x \in \mathcal{S}$.*

We assume that there exist conflicting domains in the domain set $D$. Since conflicting domains involve different target functions, one needs multiple models to capture those different input-output maps in the training data. We thus equip the learner with a *model set* $H = \{h_i\}_{i=1}^K$ parameterized by $\Theta = \{\theta_i\}_{i=1}^K$ with cardinality $K > 1$, in contrast to conventional supervised learning that only trains a global model. Intuitively, $K$ should be sufficiently large to fully resolve the conflict in the training data; this depends not only on the number of domains, but also how "severe" the conflict is among all domains. We formalize this notion by introducing the *conflict rank* of the domain set.

**Definition 2** (Conflict rank). *For a domain set $D = \{\langle P_i, c_i \rangle\}_{i=1}^N$, its* conflict rank *is defined as*

$$R(D) := \max_{\mathcal{S} \subseteq \mathcal{X}, \lambda(\mathcal{S}) > 0} \min_{x \in \mathcal{S}} \left| \left\{ c(x) \mid \langle P, c \rangle \in D, x \in \mathrm{supp}(P) \right\} \right|. \tag{1}$$

---

[1]For brevity, we omit the subscript $X$ in $P_X$ when it is clear from the context.

Note that the cardinality of a set only counts the number of its distinct elements. In brief, the conflict rank of a domain set is the maximum number of different outputs conditioned on the same input across all domains, holding for all inputs in a subset of $\mathcal{X}$ with non-zero Lebesgue measure. We will theoretically show in Sec. 3 that $R(D)$ plays an important role in the learnability of our framework.

The training data is drawn in a bilevel fashion. Concretely, we consider a domain distribution $Q$ over the domain set $D$. Each batch of the training data is drawn from a domain from $D$, where the domain itself is drawn from the domain distribution $Q$. Given the number of batches $m$ and batch size $n$, $m$ domain examples $\mathbf{d}^1, \cdots, \mathbf{d}^m$ are first i.i.d. drawn from $Q$ (a domain can recur multiple times); then, for each $i \in [m]$, $n$ inputs $\mathbf{x}^{i1}, \cdots, \mathbf{x}^{in}$ are i.i.d. drawn from the input distribution $P^i$ corresponding to $\mathbf{d}^i = \langle P^i, c^i \rangle$ and labeled by $c^i$: $\mathbf{y}^{ij} = c^i(\mathbf{x}^{ij}), \forall j \in [n]$. The key deviation from classical supervised learning is that the identities of domains are *unobservable*, thus the learner does not know from which domain each batch is drawn. An implicit assumption made here is that the examples in the same batch belong to the same domain when $n > 1$, which is reasonable since we usually do not expect the hidden contexts to change capriciously in practice (Helmbold & Long, 1994; Harries et al., 1998). Moreover, we also show that this assumption is not only reasonable but also *necessary* for our setup under some conditions (Appendix G), and theoretically and empirically validate the robustness of our method under the scenario where each batch may contain cross-domain noise (Appendix C.3 and E.1). Finally, we note that while our sampling process is akin to meta-learning, our main goal differs in that we aim to learn multiple *local* models that independently capture conflicting target functions, while meta-learning learns a *global* meta-model that enables cross-domain adaptation between related domains (Thrun & Pratt, 1998; Pentina & Lampert, 2014; Amit & Meir, 2018).

## 2.2 THE LEAF FRAMEWORK

In this section, we formulate the training objective of LEAF. Recall that our goal is to learn all target functions from potentially conflicting domains. Then, a straightforward start point is the empirical multi-task error, i.e., the average loss over all batches given a data allocation oracle:

$$\widehat{er}_{\text{MTL}}(H) = \frac{1}{m} \sum_{i=1}^{m} \frac{1}{n} \sum_{j=1}^{n} \ell \left[ g_{\text{oracle}}(i) \left( \mathbf{x}^{ij} \right), \mathbf{y}^{ij} \right], \tag{2}$$

where $g_{\text{oracle}} : [m] \to H$ determines to which model in $H$ each batch is assigned so that the conflict in training data can be eliminated. However, $g_{\text{oracle}}$ is unavailable in our setting; we thus replace it with a data-driven *allocation function* that searches over the space of allocations to resolve the conflict. In classical supervised learning, those allocations are manually designed and thus rely on the understanding of the designers on the task; by contrast, we aim to learn data allocations end-to-end without additional human intervention. We then introduce the training objective of LEAF:

**Definition 3** (LEAF training objective). *For every $i \in [m]$, let $Z^i = \{(\mathbf{x}^{ij}, \mathbf{y}^{ij})\}_{j=1}^{n}$ be the examples in the $i$-th batch. Then, the* empirical error *of LEAF is given by*

$$\widehat{er}(H) := \frac{1}{m} \sum_{i=1}^{m} \frac{1}{n} \sum_{j=1}^{n} \ell \left[ \hat{g} \left( H, Z^i \right) \left( \mathbf{x}^{ij} \right), \mathbf{y}^{ij} \right], \tag{3}$$

*where $\hat{g} : \mathcal{H}^K \times \mathcal{X}^n \times \mathcal{Y}^n \to H$ is the* empirical allocation function*; and the* expected error *is*

$$er(H) := \mathbb{E}_{\mathbf{d} = \langle P, c \rangle \sim Q} \mathbb{E}_{\mathbf{x} \sim P} \ell \left[ g(H, \mathbf{d})(\mathbf{x}), c(\mathbf{x}) \right], \tag{4}$$

*where $g : \mathcal{H}^K \times D \to H$ is the* expected allocation function*.*

Besides the empirical error $\widehat{er}(H)$, we also introduce an expected error $er(H)$ for LEAF. These definitions resemble the empirical and expected risks in classic supervised learning (Vapnik, 1999): while we only have finite examples to identify the domains and perform data allocation in practice, we also want to study to what extent this strategy differs from the *best* we can do theoretically — by giving the (expected) allocation function access to the true data distribution of each domain as in $er(H)$. This is captured by our generalization error analysis in Sec. 3 (Theorem 3).

## 2.3 ALLOCATION FUNCTION DESIGN AND OVERALL ALGORITHM

Given the objective (3), a naive implementation of LEAF is to parameterize the allocation function $\hat{g}$ with extra parameters independent of model parameters $\Theta$ and train the whole system end-to-end.

However, this design choice requires additional computational overhead and, more importantly, does not sufficiently consider the interaction between the high-level allocation function and low-level prediction models. In this section, we introduce a reinterpretation of LEAF from the angle of likelihood maximization, and derive an *analytic* form of the allocation function by drawing a connection between LEAF and a variant of the EM algorithm (Dempster et al., 1977) on conflicting data. Let $p_\Theta(Y \mid X)$ be the predictive conditional distribution of the model set $H$ parameterized by $\Theta$, where we use $X$ and $Y$ as the shorthand for $(\mathbf{x}^{11}, \cdots, \mathbf{x}^{mn})$ and $(\mathbf{y}^{11}, \cdots, \mathbf{y}^{mn})$, respectively. We consider maximizing a lower bound of the empirical conditional log-likelihood

$$\log p_\Theta(Y \mid X) \geq \sum_{i=1}^{m} \sum_{k=1}^{K} q(h_k) \sum_{j=1}^{n} \log \frac{p_\Theta(\mathbf{y}^{ij}, h_k \mid \mathbf{x}^{ij})}{q(h_k)} := F(q, \Theta), \tag{5}$$

where $q$ is a posterior distribution over $H$. By an EM-style derivation (see details in Appendix A.1), we break the optimization on the $i$-th batch into two alternating steps: in the E-step, we maximize $F(q, \Theta)$ by updating model posterior $q$; in the M-step, we maximize $F(q, \Theta)$ by updating model parameters $\Theta$ through maximizing the weighted sum of likelihoods $\sum_{k=1}^{K} q(h_k) \sum_{j=1}^{n} \log p_\Theta(\mathbf{y}^{ij} \mid \mathbf{x}^{ij}, h_k)$ given $q$. This suggests a connection between EM and LEAF: the E-step corresponds to the functionality of the allocation function that selects a model in $H$ for a given data batch, while the M-step corresponds to updating the selected model by minimizing the prediction error.

In the common practice of EM, the output of the E-step is a *multimodal* distribution over components. Nevertheless, here we expect a *unimodal* output. This is because the aim of the allocation function is to resolve the conflict via data allocation, thus conflicting examples should be deterministically assigned to different models. We enforce a unimodal E-step by assuming the posterior $q$ as a Kronecker-Delta distribution over $H$. This variant has also been explored in the literature, referred to as Viterbi EM (Brown et al., 1993; Spitkovsky et al., 2010) or hard EM (Samdani et al., 2012). We empirically observe that this choice brings consistent performance gains over standard EM, in terms of both final performance and robustness. Under the unimodality constraint, the E-step yields

$$q(h) = \delta\left(h = \arg\max_{h' \in H} \sum_{j=1}^{n} \log p_\Theta\left(\mathbf{y}^{ij} \mid \mathbf{x}^{ij}, h'\right)\right) \tag{6}$$

under a uniform model prior. This motivates a principled choice of the allocation functions:

$$\hat{g}\left(H, Z^i\right) = \arg\min_{h \in H} \sum_{j=1}^{n} \ell\left[h(\mathbf{x}^{ij}), \mathbf{y}^{ij}\right], \forall i \in [m], \tag{7a}$$

$$g(H, d_i) = \arg\min_{h \in H} \mathbb{E}_{\mathbf{x} \sim P_i} \ell\left[h(\mathbf{x}), c_i(\mathbf{x})\right], \forall i \in [N]. \tag{7b}$$

In other words, a simple implementation that selects a model from $H$ that yields the *smallest* error suffices. Compared to the naive implementation using a separately parameterized $\hat{g}$, this choice requires no extra parameters and is free from explicit bilevel optimization. We note that the connection between Eqs. (6) and (7) is not rigorous in general, with no assumptions on the forms of loss function and conditional distribution family. Fortunately, we show in Appendix A.2 that an exact equivalence can be derived in both common regression and classification settings under some natural assumptions.

**Overall algorithm.** For each data batch, LEAF consists of two phases: **(i)** for each model in $H$, evaluate its average error on the batch; **(ii)** update the model that yields the smallest error using the batch of data. We provide the pseudo-code of LEAF in Appendix B, and the implementation details for different tasks in our experiments in Appendix D.

**Convergence guarantee.** Since LEAF is derived from an EM framework, existing convergence guarantee of (generalized) EM also applies here. Notably, it is known that under fairly general conditions on the monotonicity of the likelihood with respect to posterior $q$ in each iteration, the likelihood values in (generalized) EM converge to stationary points (see e.g., Chapter 3 of McLachlan & Krishnan (2007)). Similarly, LEAF shares a similar guarantee of converging to a stationary point of the empirical error (3) under similar conditions. See Appendix A.4 for more discussion.

**Connection to latent variable modeling.** In a broad sense, LEAF is a type of latent variable modeling approach by treating the domain identity of each data pair as a (discrete) latent variable. However, to the best of our knowledge, no prior methods actually focus on our conflicting data setup, to which LEAF is tailored with several key design choices and assumptions that differ it from general latent variable modeling methods. We provide more discussion in Appendix F and G.

## 3 THEORETICAL ANALYSIS

In this section, we provide theoretical justifications of LEAF on its identifiability, learnability, and generalization error. Our theory answers the following key questions: **(i)** Is minimizing the expected error (4) sufficient for identifying conflicting domains? **(ii)** What is the minimal number of models $K$ required to achieve a low error? **(iii)** Is minimizing the empirical error (3) sufficient for minimizing the expected error, given the discrepancy induced by *both* the empirical allocation function and the empirical error minimization process compared with their expected counterparts? We assume that all target functions are realizable, i.e., contained by the hypothesis space $\mathcal{H}$. This assumption is reasonable for rich hypothesis spaces (e.g., parameterized DNNs), and helps underline the key characteristics of our problem. The proofs of all theoretical results are deferred to Appendix C.

We first analyze whether minimizing the expected error (4) with the choice of the allocation function (7b) is sufficient for identifying the conflicting domains and recovering all target functions. We give an affirmative answer to this question by characterizing the form of the optimal solution of (4).

**Theorem 1** (Identifiability of conflicting domains). *The following two propositions are equivalent:*

*(i) For the domain distribution $Q$ satisfying that $Q(d) > 0$ for every $d \in D$, the expected error of LEAF satisfies $er(H) = 0$.*

*(ii) For each domain $d = \langle P, c \rangle$ in $D$, there exists $h \in H$ such that $\mathbb{E}_{\mathbf{x} \sim P}\left[\ell[h(\mathbf{x}), c(\mathbf{x})]\right] = 0$.*

Theorem 1 suggests that minimizing the expected error naturally elicits a global optimal solution where every target function is learned accurately on the support of its corresponding input distribution, and errs only on sets with zero Lebesgue measure. We then focus on the condition under which a zero expected error can be achieved. This naturally relates to the learnability of LEAF under the framework of Probably Approximately Correct (PAC) learning (Valiant, 1984) (see detailed definitions in Appendix C.2). While learnability in the conventional PAC analysis mainly depends on the complexity of the hypothesis space (cf. Chapter 6.4 of Shalev-Shwartz & Ben-David (2014)), conflicting data incurs a new source of complexity by its *conflict rank*, which we expect can be compensated by the cardinality of the model set. We prove a necessary condition of the PAC learnability of LEAF. Since our result also applys to single-model ERM ($K = 1$), it also illustrates the infeasibility of only training a global model from a theoretical perspective.

**Theorem 2** (PAC learnability). *For a domain set $D$ with conflict rank $R(D)$ and a model set with cardinality $K$, a necessary condition of the PAC learnability of LEAF is $K \geq R(D)$.*

In other words, the cardinality of the model set needs to be at least equal to the conflict rank of the domain set to guarantee that our problem is possibly PAC-learnable. Note that this requirement is not directly related to the number of domains; it relies more on the innate structural property of the data sampled from the conflicting domains characterized by the conflict rank of the domain set.

**Remark 1.** While it is generally hard to derive a both necessary and sufficient condition of PAC learnability theoretically (which requires a sample-efficient optimization algorithm), we empirically observe that $K \geq R(D)$ is indeed an essential condition of learnability with rich hypothesis spaces implemented as parameterized DNNs. We also note that several works (Allen-Zhu et al., 2019; Du et al., 2019) have proved that over-parameterized NNs trained by SGD can achieve zero training error in polynomial time, which implies a PAC learnability as long as the conflict in the training data is removed (referred to as the non-degenerate data assumption in Allen-Zhu et al. (2019)). We leave a more rigorous analysis for future work.

The last question we are concerned about is whether the discrepancy between our empirical and expected objectives (3) (4), also known as the generalization error, can be controlled with theoretical guarantees. This question is crucial since we only have access to the empirical error estimated by the empirical examples sampled from the underlying distributions in practice, while our ultimate goal is to minimize the expected error. While conventional supervised learning enjoys statistical guarantees on the upper bound of the generalization error induced by the single-model ERM process (Vapnik, 1999), LEAF introduces another source of generalization error by the discrepancy between the empirical and expected *allocation functions*. Therefore, upper-bounding the generalization error of LEAF is important and non-trivial. We present Theorem 3 that shows an upper bound of the generalization error when the data in each batch is noiseless, i.e., sampled from the same domain, and prove a more general result in Appendix C.3 under the case where each batch is itself a mixture of the data from the sampled domain and other domains with a given mixing ratio.

**Theorem 3** (Generalization error upper bound). *For any $\delta \in (0, 1]$, the following inequality holds uniformly for all model sets $H \in \mathcal{H}^K$ with probability at least $1 - \delta$:*

$$er(H) \le \widehat{er}(H) + \underbrace{\sqrt{\frac{\mathscr{D}(\bar{\mathcal{S}})\left(\ln 2m/\mathscr{D}(\bar{\mathcal{S}}) + 1\right) - \ln \delta/24}{m}} + \frac{1}{m}}_{\text{domain estimation error}} \tag{8a}$$

$$+ \underbrace{\sum_{k=1}^{N}\left(\frac{m_k}{m}\sqrt{\frac{\mathscr{D}(\mathcal{S})\left(\ln 2m_k n/\mathscr{D}(\mathcal{S}) + 1\right) - \ln \delta/12N}{m_k n}} + \frac{1}{mn}\right)}_{\text{instance estimation error}} \tag{8b}$$

$$+ \underbrace{2\sqrt{\frac{\mathscr{D}(\mathcal{S})\left(\ln 2n/\mathscr{D}(\mathcal{S}) + 1\right) - \ln \delta/24m}{n}} + \frac{2}{n}}_{\text{allocation estimation error}}, \tag{8c}$$

*where $\bar{\mathcal{S}} = \{\langle P, c\rangle \mapsto \min_{h \in H} \mathbb{E}_{\mathbf{x} \sim P} \ell\left[h(\mathbf{x}; \Theta), c(\mathbf{x})\right]\}$ is the function set of the expected domain-wise error, $\mathcal{S} = \{(x, y) \mapsto \ell\left[h(x; \Theta), y\right]\}$ is the function set of the instance-wise error, $m_k = \sum_{i=1}^{m} \mathbb{1}\left(\mathbf{d}^i = d_k\right), k \in [N]$ is the count of the $k$-th domain in $D$ during the sampling process, and $\mathscr{D}(\cdot)$ is the VC-dimension (Vapnik & Chervonenkis, 1971) for some set of real functions.*

Theorem 3 indicates that the expected error $er(H)$ is bounded by the empirical error $\widehat{er}(H)$ plus three terms. The *domain estimation error* term (8a) is derived by bounding the discrepancy between the empirical and expected domain distributions; it converges to zero when the number of batches $m \to \infty$. The *instance estimation error* term (8b) is derived by bounding the discrepancy between the empirical and expected data distributions; it converges to zero when $m \to \infty$ *or* the batch size $n \to \infty$, which captures the effect of the recurrence of domains during sampling. The *allocation estimation error* term (8c) is derived by bounding the discrepancy between the empirical and the expected allocation functions; it converge to zero when $n \to \infty$. In practice, we find a very small $n$ often suffices for controlling the generalization error. We posit that this is due to that the domains in our experiments are rather diverse, thus reducing the difficulty of discriminating between them.

**Comparison with existing bounds.** We compare the general form of our bound (8) with existing bounds in classical supervised learning and meta-learning. In general, supervised learning bounds (Vapnik, 1999; McAllester, 1999) contain an instance-level error term akin to (8b), while meta-learning bounds (Pentina & Lampert, 2014; Amit & Meir, 2018) further contain a domain-level error term akin to (8a) (with no explicit treatment of domain recurrences). Yet, classical supervised learning only considers a single domain or multiple domains with known example-domain correspondances; meta-learning treats each batch as a *new* task and learns a *global* meta-model for adaptation. Thus, none of these bounds contain an explicit model inference error term as (8c).

**Remark 2.** While our bound applies VC-dimension as the complexity measure, extensions to other data-dependent complexity measures such as the Rademacher complexity (Bartlett & Mendelson, 2002; Koltchinskii & Panchenko, 2000) are straightforward.

## 4 EXPERIMENTAL EVALUATION

In this section, we evaluate LEAF on two common supervised learning settings of regression and classification that involve conflicting data. Our experiments are designed to (i) assess the efficacy of LEAF on eliminating the conflict in the training data, (ii) showcase the applicability of LEAF in three practical classification scenarios that involve multiple paralleled, hierarchical, or opposite input-output relations in different domains, and (iii) compare LEAF with task-specific approaches.

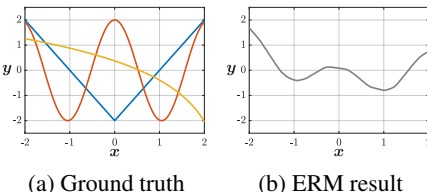

(a) Ground truth    (b) ERM result

Figure 1: The ground truth of the regression task and the result of an ERM learner.

### 4.1 REGRESSION

We start with an illustrative regression problem in which the examples are simultaneously sampled from three heterogeneous functions with additive Gaussian noise. The ground truth of these functions

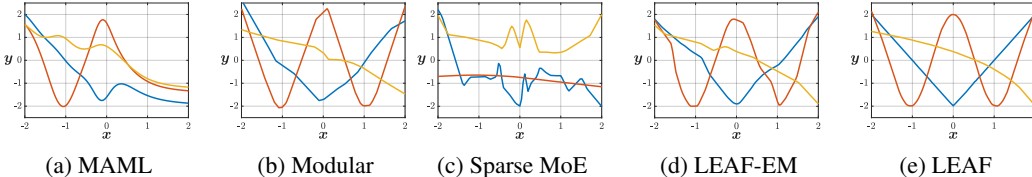

| (a) MAML | (b) Modular | (c) Sparse MoE | (d) LEAF-EM | (e) LEAF |

Figure 2: Results on the regression task where the examples are simultaneously sampled from three heterogeneous functions. LEAF is the only method that smoothly recovers all functions.

is shown in Figure 1a. In this task, a vanilla ERM learner will trivially output the mean of all functions as shown in Figure 1b. Each prediction model is instantiated by a neural network regressor with ReLU nonlinearities trained by Adam (Kingma & Ba, 2015). More details are in Appendix D.1.

**Baselines.** We compare LEAF with two types of baselines: (i) meta-learning that learns a global meta-model that can be rapidly fine-tuned to different functions, and (ii) Mixture-of-Experts (MoE) that learns a set of expert networks and a parameterized gating network that fuses the outputs of experts. For (i), we use *MAML* (Finn et al., 2017), a widely-used gradient-based meta-learning approach, and *Modular* meta-learning (Alet et al., 2018) that extends MAML with multiple modules. For (ii), we use *Sparse MoE* (Shazeer et al., 2017) that achieves state-of-the-art results on language modeling tasks; we augment the input of the gating network with the ground-truth outputs to make it sensible of the conflict, and enforce that only one expert is selected by the gating function for each example. We also compare a variant of our method, *LEAF-EM* that uses standard EM without the unimodal constraint. More details on the baselines are in Appendix D.1.1.

**Evaluation protocol.** For meta-learning methods, we use additional examples for the learned model to adapt to each function and plot the fitted curves after adaptation. For other methods, we directly plot the fitted curves of all models or experts and compare the outputs with the ground-truth functions.

**Results.** As shown in Figure 2, LEAF is the only method that smoothly recovers all functions. Meta-learning methods do not produce accurate predictions in fine-grained details due to their nature of few-shot fine-tuning. Sparse MoE completely fails in this task; we posit that this is because the gating network in MoE is separately parameterized and trained end-to-end, leading to optimization difficulty on conflicting data. By contrast, the analytic allocation function relieves the burden of learning additional gating parameters and leads to better results. Finally, LEAF-EM performs worse than LEAF, showing the advantage of using the unimodality constraint.

## 4.2 CLASSIFICATION

To evaluate LEAF in more practical scenarios, we consider three classification tasks that involve paralleled, hierarchical, and opposite input-output relations across the domains. These settings reflect real-world scenarios with different types of hidden contexts, as we detail below:

- **Paralleled relations.** Data from different domains has paralleled labels that represent independent attributes of the same input, such as "red" and "sphere". This is common for uncurated data, e.g., web data. For this setting, we use two datasets. (1) *Colored MNIST*, a variant of MNIST where each digit is colored and assigned with an additional color label, resulting in two domains: *Digit* and *Color*. (2) *Fashion Product Images* (Aggarwal, 2019), a multi-attribute product classification dataset that involves three domains: *Gender*, *Category* and *Color*, as shown in Figure 3a.

- **Hierarchical relations.** Data from different domains has labels with an innate hierarchy, such as "person" and "scuba diver". This is common for different datasets with different labeling purposes or different levels of expertise of the human annotators. For this setting, we use the *CIFAR-100* dataset (Krizhevsky & Hinton, 2009), a widely-used image recognition dataset that consists of 100 classes with fine labels subsumed by 20 superclasses with coarse labels, resulting in two domains: *Superclass* and *Class* with a hierarchical label space structure, as shown in Figure 3b.

- **Opposite relations.** Data from different domains has completely opposite labels, such as "like" and "dislike". This is often due to preferences or sentiments of human users. For this setting, we re-label the *Fashion Product Images* dataset, simulating a preference-dependent scenario with two domains: *Male* and *Female*, as shown in Figure 3c. The *Male* domain treats all products with the "male" attribute as positive and other products as negative, while the *Female* domain treats all products with the "female" attribute as positive and other products negative.

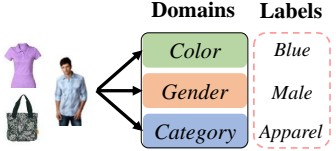
(a) Paralleled relations: *Fashion Product Images*

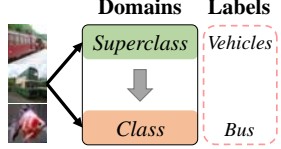
(b) Hierarchical relations: *CIFAR-100*

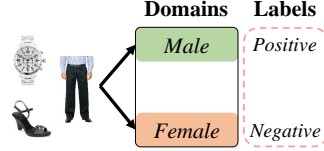
(c) Opposite relations: *Fashion Product Images* with preferences

Figure 3: Classification datasets with paralleled, hierarchical, and opposite input-output relations.

**Baselines.** (i) *ERM-top $k$*: we directly use the top-$k$ outputs of ERM. (ii) Semi-supervised multi-label learning: we alternatively interpret learning from conflicting classification data as multi-label learning (Zhang & Zhou, 2014) *with missing labels* (see Appendix D.2.1 for more discussion); in light of this, we adopt two representative methods, *PseudoLabel* (Lee, 2013) and *LabelProp* (Iscen et al., 2019), to iteratively fill up the missing labels during training. (iii) *Sparse MoE*: the same baseline as in regression, with each expert instantiated as a neural network classifier. Moreover, we consider two baselines with additional label set or domain index information. (iv) *MLL oracle*: multi-label learning where we provide the full label set for each example. (v) *MTL oracle*: multi-task learning where we provide the ground-truth domain index for each example. As in regression, we also evaluate the *LEAF-EM* variant of our method. More details of datasets and baselines are in Appendix D.2.

**Evaluation protocol.** For LEAF and other methods that use multiple low-level models, we report the errors based on the low-level model that yields the smallest error for each domain: given domain $d$ with test examples $\{(x_i, y_i)\}_{i=1}^{n_d}$, the error on $d$ is defined as $\min_{h \in H} \sum_{i=1}^{n_d} \mathbb{1}[h(x_i) \neq y_i]$, which is an unbiased estimator of the expected error (4) by taking the loss function $\ell$ to be the 0-1 loss. For methods with no multi-model architecture, we compare their output label sets with the ground-truth label set for each test example and report the error on each domain. Note that multi-label learning methods do not apply to the opposite relations setting and we thus omit their results.

**Results.** As shown in Tables 1 and 2, LEAF significantly outperforms all baselines in all three tasks, yielding smaller prediction errors on all domains. Even compared with fully supervised learning methods with label set or domain index oracles, LEAF performs competitively. Once again, we observe that LEAF consistently surpasses the *LEAF-EM* variant, suggesting the benefits of a unimodal E-step.

**Visualization of low-level feature spaces.** To complement the quantitative results, we visualize the features learned by the low-level models of LEAF on the paralleled relation setting on *Fashion Product Images*. As depicted in Figure 4, the low-level models indeed learn feature spaces that separately capture the semantics of different domains.

Table 1: Results on the classification task with opposite input-output relations. We report prediction errors on both domains, *Male* and *Female*.

| Methods | Error (%) | |
| | *Male* | *Female* |
| --- | --- | --- |
| ERM | 49.50 | 50.50 |
| Sparse MoE | 82.18 | 32.43 |
| LEAF-EM | 22.10 | 81.85 |
| **LEAF** | **7.40** | **9.65** |
| MTL oracle | 7.14 | 6.53 |

### 4.3 ADDITIONAL EXPERIMENTS AND RESULTS

We also conduct additional experiments to demonstrate the efficacy and robustness of LEAF and study the impact of the number of low-level models and parameters. Due to space limit, these experiments and results are deferred to Appendix E. Here we provide a list of them and refer the interested readers to the corresponding sections in the appendix. **(i)** In E.1, we evaluate the **robustness** of LEAF under both the cross- and in-domain noise; **(ii)** in E.2, we evaluate LEAF on a real-world **multi-dimensional regression** task; **(iii)** in E.3, we study the **impact of the number of low-level models** and sampling parameters on LEAF; **(iv)** in E.4, we report the training and inference **time cost** of LEAF and baselines in the classification task; **(v)** in E.5, we provide **additional visualization** on the iteration process and the low-level feature spaces learned by LEAF models.

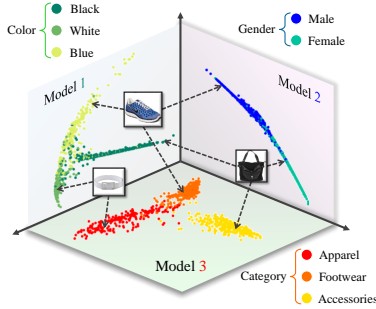

Figure 4: Visualization of different low-level feature spaces learned by LEAF on *Fashion Product Images*.

Table 2: Results on the classification tasks with paralleled and hierarchical input-output relations. We report prediction errors on *Digit* and *Color* domains of *Colored MNIST*, *Gender*, *Category* and *Color* domains of *Fashion Product Images*, and *Superclass* and *Class* domains of *CIFAR-100*.

| Methods | Colored MNIST | | Fashion Product Images | | | CIFAR-100 | |
|---|---|---|---|---|---|---|---|
| | Error (%) | | Error (%) | | | Error (%) | |
| | Digit | Color | Gender | Category | Color | Superclass | Class |
| ERM-top $k$ | 3.04 | 0.39 | 22.80 | 18.02 | 33.15 | 27.35 | 31.20 |
| PseudoLabel | 7.47 | 10.08 | 33.69 | 20.04 | 34.06 | 28.46 | 38.96 |
| LabelProp | 11.52 | 13.57 | 14.59 | 50.43 | 64.48 | 66.62 | 45.17 |
| Sparse MoE | 15.70 | 25.40 | 37.13 | 27.76 | 44.28 | 45.63 | 51.15 |
| LEAF-EM | 6.83 | 4.76 | 26.67 | 6.01 | 18.70 | 32.71 | 33.83 |
| **LEAF** | **1.70** | **0.03** | **7.87** | **1.93** | **12.85** | **21.40** | **25.05** |
| MLL oracle | 1.02 | 0.00 | 8.46 | 1.17 | 9.45 | 22.08 | 26.29 |
| MTL oracle | 1.20 | 0.00 | 7.14 | 1.90 | 11.04 | 21.11 | 25.08 |

## 5 RELATED WORK

Early work on supervised learning with the presence of hidden contexts (Schlimmer & Granger, 1986; Helmbold & Long, 1994; Widmer & Kubat, 1993) adopts the notion of context to model concept drifting. Most of them consider an *online learning* scenario and aim to track the changing contexts in a datastream. Harries et al. (1998) propose to extract hidden contexts from the offline data using contextual clustering, but their method requires additional contextual clues and does not scale to modern machine learning datasets. More recent work explores to leverage the available contextual information to facilitate learning in various scenarios, such as question answering (Choi et al., 2018), image and scene understanding (Shotton et al., 2009; Chen et al., 2015), retrieval (Zhao et al., 2018), and recommendation systems (Pazzani & Billsus, 2007). A series of works either implicitly or explicitly model the contexts in different data distributions to improve the *adaptability* of a global meta-model (Yao et al., 2019; Zintgraf et al., 2019; Rakelly et al., 2019; Harrison et al., 2019). Several works on similarity learning investigate the case where the similarity between images depends on different latent attributes (Amid & Ukkonen, 2015; Veit et al., 2017; Nigam et al., 2019), requiring image triplets as the inputs. Su et al. (2020) study a multi-task learning problem where the goal of the learner is to learn from online examples without task indices, but with a different learning objective from ours and no theoretical justification.

Extensive literature has explored the collaboration of multiple models or modules in completing one or multiple tasks (Doya et al., 2002; Andreas et al., 2016; Alet et al., 2018; Meyerson & Miikkulainen, 2019; Yang et al., 2020; Yuksel et al., 2012; Shazeer et al., 2017). The difference between these methods and LEAF is that the multi-model architecture of LEAF is driven by the inherent conflict in conflicting data, and we only allow a single low-level model to be invoked for a particular batch. We provide more discussion on the related work in Appendix F.

## 6 LIMITATIONS AND DISCUSSION

In this work, we formulate a novel supervised learning setting of learning from conflicting data, which captures the key difficulty in training on the data with multiple hidden contexts. We hope that our work can serve as a stepping stone in the pursuit of more general and automatic learning frameworks, and list some limitations of the current work that are worth exploring in the future:

**Principled methods for handling noisy data.** While we have empirically assessed the robustness of LEAF under both cross-domain and in-domain noise (Appendix E.1), it is promising to combine LEAF with robust learning approaches (Zhang & Sabuncu, 2018; Song et al., 2022) to develop more principled methods for distinguishing hidden contexts from data noise.

**Determining the number of prediction models.** A critical problem in LEAF is to determine the number of low-level models. In Appendix E.3, we discuss several heuristics for this problem and provide primary experimental results. Nonetheless, developing theoretically grounded approaches for choosing or adapting the number of prediction models is an exciting future direction.

ETHICS STATEMENT

In a broad sense, the goal of this work is to contribute to the development of more general and automatic machine learning frameworks. Therefore, like other methods that also aim to reduce the human labor in machine learning, our work may induce negative societal impacts due to the concerns of the robustness and explainability of our models. While devising more robust and explainable machine learning models has been considered as paralleled research fields, in this work we provide primary robustness assessments of LEAF in Appendix E.1 and show that our framework does not exhibit a more severe robustness downside compared with existing deep learning models that are widely used today. We also note that our method can benefit from future advances in robust and explainable machine learning research. On the other hand, by reducing the need of explicit context identifiers during training, our method may facilitate machine learning research in privacy-sensitive scenarios where contexts are not available due to privacy constraints.

REPRODUCIBILITY STATEMENT

This paper includes both theoretical analysis and experimental results. For theoretical results, we state all assumptions and discuss the reasons for introducing them before presenting the results, as in Sec. 3. Complete proofs of all theoretical results and relevant definitions are included in Appendix C. For experimental results, we specify the training details (e.g., data splits, hyperparameters, network architectures) in Appendix D and provide the source code in the supplementary material.

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

# APPENDIX

## A  CONNECTION BETWEEN LEAF AND EM

### A.1  DERIVATION DETAILS OF E-STEP AND M-STEP OBJECTIVES

In this section, we provide the derivation details of the E-step and M-step objectives stated in Sec. 2.3. As we have mentioned in the main text, we reinterpret LEAF through the lens of likelihood maximization and seek to maximize the conditional log-likelihood

$$\log p_\Theta(Y \,|\, X) = \sum_{i=1}^{m} \sum_{j=1}^{n} \log \sum_{k=1}^{K} p_\Theta(\mathbf{y}^{ij}, h_k \,|\, \mathbf{x}^{ij}), \tag{9}$$

where $X = (\mathbf{x}^{11}, \cdots, \mathbf{x}^{mn})$ and $Y = (\mathbf{y}^{11}, \cdots, \mathbf{y}^{mn})$. We then show that a lower bound of the conditional likelihood (9) is given by

$$
\begin{aligned}
\log p_\Theta(Y \,|\, X) &= \sum_{i=1}^{m} \sum_{j=1}^{n} \log \sum_{k=1}^{K} q(h_k) \frac{p_\Theta(\mathbf{y}^{ij}, h_k \,|\, \mathbf{x}^{ij})}{q(h_k)} \\
&\geq \sum_{i=1}^{m} \sum_{j=1}^{n} \sum_{k=1}^{K} q(h_k) \log \frac{p_\Theta(\mathbf{y}^{ij}, h_k \,|\, \mathbf{x}^{ij})}{q(h_k)} \\
&= \sum_{i=1}^{m} \sum_{j=1}^{n} \sum_{k=1}^{K} q(h_k) \log \left[ p(h_k \,|\, \mathbf{x}^{ij}) p_\Theta(\mathbf{y}^{ij} \,|\, \mathbf{x}^{ij}, h_k) \right] \\
&\quad - \sum_{i=1}^{m} \sum_{j=1}^{n} \sum_{k=1}^{K} q(h_k) \log q(h_k), \\
&:= F(q, \Theta)
\end{aligned}
\tag{10}
$$

where $q$ is an arbitrary posterior distribution over the model set $H$, and the second step in the derivation follows from Jensen's Inequality. The EM algorithm can then be considered as performing coordinate ascent alternating over $q$ and $\Theta$, corresponding to the E-step and the M-step, respectively. Using Viterbi EM (Brown et al., 1993; Spitkovsky et al., 2010; Samdani et al., 2012), we assume that $q$ is a Kronecker-Delta distribution, which outputs a non-zero probability mass only on a single point. Due to the unimodularity of simplex constraints (Samdani et al., 2012), this results in an analytic $q$:

$$
\begin{aligned}
q(h) &= \delta \left( h = \arg\max_{h' \in H} \log p_\Theta(\mathbf{y}^{ij}, h' \,|\, \mathbf{x}^{ij}) \right) \\
&= \delta \left( h = \arg\max_{h' \in H} \log p(h' \,|\, \mathbf{x}^{ij}) p_\Theta(\mathbf{y}^{ij} \,|\, \mathbf{x}^{ij}, h') \right).
\end{aligned}
\tag{11}
$$

By assuming a uniform model prior $p(h \,|\, \mathbf{x}^{ij})$, we have

$$q(h) = \delta \left( h = \arg\max_{h' \in H} \log p_\Theta(\mathbf{y}^{ij} \,|\, \mathbf{x}^{ij}, h') \right). \tag{12}$$

Given the posterior $q$ and the uniform model prior, in the M-step we maximize

$$F(\Theta) = \sum_{i=1}^{m} \sum_{j=1}^{n} \sum_{k=1}^{K} q(h_k) \log p_\Theta(\mathbf{y}^{ij} \,|\, \mathbf{x}^{ij}, h_k). \tag{13}$$

**Adaptation to the batch setting.**  The model posterior given by Equation (12) is estimated based on each data pair $(\mathbf{x}^{ij}, \mathbf{y}^{ij})$. Note that in each data batch, different data pairs are i.i.d. drawn from the sampled domain. We can leverage this fact to integrate the estimations of the model posterior obtained by different empirical examples.

For example, consider the $i$-th batch $Z^i = \{(\mathbf{x}^{ij}, \mathbf{y}^{ij})\}_{j=1}^{n}$. Since different data pairs $(\mathbf{x}^{ij}, \mathbf{y}^{ij})$, $j \in [n]$ are i.i.d., their corresponding posteriors $q(h)$ are also identical. Hence, given the data batch

$Z^i = \{(\mathbf{x}^{ij}, \mathbf{y}^{ij})\}_{j=1}^n$, we have

$$q(h) = q(h \mid Z^i) = q(h \mid \mathbf{x}^{ij}, \mathbf{y}^{ij}) \quad \forall j \in [n]$$
$$= \delta\left( h = \arg\max_{h' \in H} \sum_{j=1}^n \log p_\Theta(\mathbf{y}^{ij} \mid \mathbf{x}^{ij}, h') \right). \tag{14}$$

Similarly, we can rewrite the overall M-step objective (13):

$$F(\Theta) = \sum_{i=1}^m \sum_{j=1}^n \sum_{k=1}^K q(h_k \mid \mathbf{x}^{ij}, \mathbf{y}^{ij}) \log p_\Theta(\mathbf{y}^{ij} \mid \mathbf{x}^{ij}, h_k)$$
$$= \sum_{i=1}^m \sum_{k=1}^K \sum_{j=1}^n q(h_k \mid Z^i) \log p_\Theta(\mathbf{y}^{ij} \mid \mathbf{x}^{ij}, h_k) \tag{15}$$
$$= \sum_{i=1}^m \sum_{k=1}^K q(h_k \mid Z^i) \sum_{j=1}^n \log p_\Theta(\mathbf{y}^{ij} \mid \mathbf{x}^{ij}, h_k).$$

We can then write the E-step and the M-step objectives in the $i$-th batch as:

- E-step: set $q(h) = \delta\left( h = \arg\max_{h' \in H} \sum_{j=1}^n \log p_\Theta(\mathbf{y}^{ij} \mid \mathbf{x}^{ij}, h') \right).$

- M-step: given $q$, maximize

$$F^i(\Theta) = \sum_{k=1}^K q(h_k) \sum_{j=1}^n \log p_\Theta(\mathbf{y}^{ij} \mid \mathbf{x}^{ij}, h_k). \tag{16}$$

## A.2 DISCUSSION ON THE ALLOCATION FUNCTION AND MODEL POSTERIOR ESTIMATION

In Sec. 2.3, we motivate an analytic form of the allocation function using an EM-based maximum likelihood formulation. However, the exact equivalence between the E-step in EM and the allocation function has not been rigorously proved since it relies on the exact forms of the loss function $\ell$ and the conditional likelihoods $p_\Theta(\mathbf{y}^{ij} \mid \mathbf{x}^{ij}, h)$. As a complement, in this section we consider standard regression and classification settings, and show that the exact equivalence between estimating the model posterior and the allocation function can be obtained under some natural assumptions on the loss function and conditional distribution families.

### A.2.1 REGRESSION WITH ISOTROPIC GAUSSIAN JOINT LIKELIHOOD

Consider a regression task where we assume that for each example $(\mathbf{x}^{ij}, \mathbf{y}^{ij})$, $i \in [m]$, $j \in [n]$, the joint distribution of conditional likelihoods $p_\Theta(\mathbf{y}^{ij} \mid \mathbf{x}^{ij}, h_k)$, $k \in [K]$ conforms to an isotropic Gaussian $\mathcal{N}(\boldsymbol{\mu}_\Theta^{ij}, \sigma^2 \boldsymbol{I}_K)$ with mean $\boldsymbol{\mu}_\Theta^{ij} = (h_1(\mathbf{x}^{ij}), h_2(\mathbf{x}^{ij}), \cdots, h_K(\mathbf{x}^{ij}))$ and variance $\sigma^2 \boldsymbol{I}_K$, where $\boldsymbol{I}_K$ is a $K \times K$ identity matrix. We then write Equation (14) as

$$q(h) = \delta\left( h = \arg\max_{h' \in H} \sum_{j=1}^n \log p_\Theta(\mathbf{y}^{ij} \mid \mathbf{x}^{ij}, h') \right)$$
$$= \delta\left( h = \arg\max_{h' \in H} \sum_{j=1}^n \log \frac{1}{\sqrt{2\pi}\sigma} \exp\left[ -\frac{(\mathbf{y}^{ij} - h'(\mathbf{x}^{ij}))^2}{2\sigma^2} \right] \right)$$
$$= \delta\left( h = \arg\max_{h' \in H} \log \frac{1}{(2\pi)^{\frac{n}{2}} \sigma^n} \exp\left[ -\frac{\sum_{j=1}^n (\mathbf{y}^{ij} - h'(\mathbf{x}^{ij}))^2}{2\sigma^2} \right] \right) \tag{17}$$
$$= \delta\left( h = \arg\min_{h' \in H} \sum_{j=1}^n (\mathbf{y}^{ij} - h'(\mathbf{x}^{ij}))^2 \right).$$

This amounts to the empirical allocation function (7a) with squared loss $\ell(y, y') = (y - y')^2$.

### A.2.2 Classification with independent categorical likelihoods

Consider an $L$-way classification task where we assume that for each example $(\mathbf{x}^{ij}, \mathbf{y}^{ij})$, $i \in [m]$, $j \in [n]$, the distributions of conditional likelihoods $p_\Theta(\mathbf{y}^{ij} \,|\, \mathbf{x}^{ij}, h_k)$, $k \in [K]$ conform to independent categorical distributions with parameters $(\boldsymbol{\lambda}^{ij}_{\Theta,1}, \boldsymbol{\lambda}^{ij}_{\Theta,2}, \cdots, \boldsymbol{\lambda}^{ij}_{\Theta,K}) = (h_1(\mathbf{x}^{ij}), h_2(\mathbf{x}^{ij}), \cdots, h_K(\mathbf{x}^{ij}))$, where each $h_k(\mathbf{x}^{ij}) = (h_{k1}(\mathbf{x}^{ij}), \cdots, h_{kL}(\mathbf{x}^{ij}))$, $k \in [K]$ is an $L$-dimensional, non-negative vector that sums up to 1, representing the probability of each category. We apply an one-hot encoding to the labels and let each $\mathbf{y}^{ij} = (y^{ij}_1, \cdots, y^{ij}_L) \in \{0, 1\}^L$ with $\sum_{l=1}^L y^{ij}_l = 1$. We then write Equation (14) as

$$
\begin{aligned}
q(h) &= \delta\bigg( h = \arg\max_{h_k \in H} \sum_{j=1}^n \log p_\Theta(\mathbf{y}^{ij} \,|\, \mathbf{x}^{ij}, h_k) \bigg) \\
&= \delta\bigg( h = \arg\max_{h_k \in H} \sum_{j=1}^n \log \prod_{l=1}^L h_{kl}(\mathbf{x}^{ij})^{y^{ij}_l} \bigg) \\
&= \delta\bigg( h = \arg\max_{h_k \in H} \sum_{j=1}^n \sum_{l=1}^L y^{ij}_l \log h_{kl}(\mathbf{x}^{ij}) \bigg) \\
&= \delta\bigg( h = -\arg\min_{h_k \in H} \sum_{j=1}^n \sum_{l=1}^L y^{ij}_l \log h_{kl}(\mathbf{x}^{ij}) \bigg).
\end{aligned}
\tag{18}
$$

This amounts to the empirical allocation function (7a) with cross-entropy loss $\ell(y, y') = -\sum_{l=1}^L y_l \log y'_l$.

### A.3 LEAF with standard EM

In this section, we discuss an alternative choice of a (stochastic) allocation function using standard EM. Following the derivation in Sec. A.1, we can write the E-step and M-step objectives in the $i$-th batch as:

- E-step: set $q(h) = p(h \,|\, Z^i)$.

- M-step: given $q$, maximize

$$
F^i(\Theta) = \sum_{k=1}^K q(h_k) \sum_{j=1}^n \log p_\Theta(\mathbf{y}^{ij} \,|\, \mathbf{x}^{ij}, h_k).
\tag{19}
$$

The difference is that in standard EM, the maximization in the E-step is done by setting $q$ as the posterior $p(h \,|\, Z^i)$. By Bayes' theorem, we have

$$
\begin{aligned}
p(h \,|\, Z^i) &= \frac{p(h)\, p(Z^i \,|\, h)}{p(Z^i)} \\
&= \frac{p(h) \prod_{j=1}^n p(\mathbf{x}^{ij} \,|\, h) \prod_{j=1}^n p_\Theta(\mathbf{y}^{ij} \,|\, \mathbf{x}^{ij}, h)}{p(Z^i)} \\
&= C \prod_{j=1}^n p_\Theta(\mathbf{y}^{ij} \,|\, \mathbf{x}^{ij}, h),
\end{aligned}
\tag{20}
$$

where $C$ can be viewed as a normalization constant. As in Sec. A.2, in the sequel we provide our implementation of this LEAF variant in both regression and classification settings.

### A.3.1 Regression with isotropic Gaussian joint likelihood

We consider the same regression setting as in Sec. A.2.1, where the joint distribution of conditional likelihoods $p_\Theta(\mathbf{y}^{ij} \,|\, \mathbf{x}^{ij}, h_k), k \in [K]$ conforms to an isotropic Gaussian. We then write Equation (20)

as

$$p(h \mid Z^i) = C \prod_{j=1}^{n} p_\Theta(\mathbf{y}^{ij} \mid \mathbf{x}^{ij}, h)$$

$$= C \frac{1}{(2\pi)^{\frac{n}{2}} \sigma^n} \exp\left[ -\frac{\sum_{j=1}^{n} \left(\mathbf{y}^{ij} - h(\mathbf{x}^{ij})\right)^2}{2\sigma^2} \right]. \tag{21}$$

This gives the normalized model posteriors

$$p(h_k \mid Z^i) = \frac{p(h_k \mid Z^i)}{\sum_{m=1}^{K} p(h_m \mid Z^i)}$$

$$= \frac{\exp\left[ -\frac{\sum_{j=1}^{n} \left(\mathbf{y}^{ij} - h_k\left(\mathbf{x}^{ij}\right)\right)^2}{2\sigma^2} \right]}{\sum_{m=1}^{K} \exp\left[ -\frac{\sum_{j=1}^{n} (\mathbf{y}^{ij} - h_m(\mathbf{x}^{ij}))^2}{2\sigma^2} \right]}, \ \forall k \in [K]. \tag{22}$$

In other words, each model in the model set is sampled according to a softmax probability determined by their negative squared losses on the data batch and a variance parameter $2\sigma^2$, which reflects the noise of the regression outputs. Since $\sigma^2$ is generally unknown, in practice we treat this as an additional hyperparameter (details are in Sec. D).

### A.3.2 CLASSIFICATION WITH INDEPENDENT CATEGORICAL LIKELIHOODS

We consider the same classification setting as in Sec. A.2.2, where the distributions of conditional likelihoods $p_\Theta(\mathbf{y}^{ij} \mid \mathbf{x}^{ij}, h_k), k \in [K]$ conform to independent categorical distributions. We then write Equation (20) as

$$p(h \mid Z^i) = C \prod_{j=1}^{n} p_\Theta(\mathbf{y}^{ij} \mid \mathbf{x}^{ij}, h)$$

$$= C \prod_{j=1}^{n} \prod_{l=1}^{L} h_{kl}(\mathbf{x}^{ij})^{y_l^{ij}}. \tag{23}$$

This gives the normalized posteriors

$$p(h_k \mid Z^i) = \frac{p(h_k \mid Z^i)}{\sum_{m=1}^{K} p(h_m \mid Z^i)}$$

$$= \frac{\exp\left[ y_l^{ij} \log h_{kl}(\mathbf{x}^{ij}) \right]}{\sum_{m=1}^{K} \exp\left[ y_l^{ij} \log h_{ml}(\mathbf{x}^{ij}) \right]}, \ \forall k \in [K]. \tag{24}$$

In other words, each model in the model set is sampled according to a softmax probability determined by their negative cross-entropy losses on the data batch.

### A.4 CONVERGENCE GUARANTEE OF LEAF

In this section, we discussion the convergence guarantee of LEAF based on existing convergence guarantee of (generalized) EM. It has been shown that under fairly general conditions on the monotonicity of the likelihood with respect to posterior $q$ in each iteration, the likelihood values in (generalized) EM converge to stationary points with zero gradients (McLachlan & Krishnan, 2007). On the other hand, in the analysis above, we show that for the setting of regression with squared loss and classification with cross-entropy loss, LEAF is essentially equivalent to likelihood maximization using a type of generalized EM method. These two facts then indicate that LEAF can converge to a stationary point of the empirical error (3) under the monotonicity condition that each iteration decreases the empirical error (3) by improving the allocations. In practice, this monotonicity condition may not strictly hold if we use parameterized DNNs as the low-level models of LEAF, due to the non-convexity of the training objective. However, empirically we found that LEAF always converge stably in all of our experiments.

---

**Algorithm 1** LEAF

---

**Require:** Model set $H = \{h_k\}_{k=1}^K$ parameterized by $\Theta = \{\theta_k\}_{k=1}^K$, optimizer OPT, the number of batches $m$, batch size $n$, meta-batch size $B$.

1: **for** $i = 1, 2, \cdots, m$ batches **do**
2:      Sample the examples $\{(\mathbf{x}^{ij}, \mathbf{y}^{ij}\}_{j=1}^n$.
3:      Select a model from the model set: $\hat{h}^* \in \arg\min_{h \in H} \sum_{j=1}^n \ell\left[h(\mathbf{x}^{ij}), \mathbf{y}^{ij}\right]$.
4:      Compute the gradients $\text{grad}^i \leftarrow \nabla_\Theta \frac{1}{n} \sum_{j=1}^n \ell\left[\hat{h}^*(\mathbf{x}^{ij}), \mathbf{y}^{ij}\right]$.
5:      **if** $i \equiv 0 \,(\text{mod } B)$ **or** $i = m$ **then**
6:          Average the gradients over $B$ batches: $\text{grad} \leftarrow \frac{1}{B} \sum_{p=0}^{B-1} \text{grad}^{i-p}$.
7:          Update model parameters $\Theta \leftarrow \text{OPT}(\Theta, \text{grad})$.
8:      **end if**
9: **end for**

---

## B   Algorithm pseudo-code

We provide the pseudo-code of LEAF in Algorithm 1. In our implementation, we introduce a meta-batch size hyperparameter $B$ and average the updates in each $B$ meta-batches. This is similar to the mini-batch training in standard supervised learning in order to obtain more stable parameter updates and improve the robustness against the noise during optimization.

## C   Proofs of theoretical results

In this section, we provide the full proofs of Theorems 1, 2 and prove a generalized result of Theorem 3 in the main text. For better exposition, we restate each theorem before its proof.

### C.1   Proof of Theorem 1

**Theorem 1** (Identifiability of conflicting domains)**.** *The following two propositions are equivalent:*

*(i) For the domain distribution $Q$ satisfying that $Q(d) > 0$ for every $d \in D$, the expected error of LEAF satisfies $er(H) = 0$.*

*(ii) For each domain $d = \langle P, c \rangle$ in $D$, there exists $h \in H$ such that $\mathbb{E}_{\mathbf{x} \sim P}\ell\left[h(\mathbf{x}), c(\mathbf{x})\right] = 0$.*

*Proof.* Recall the definition of the expected error $er(H)$ (4) under the expected allocation function (7b):
$$er(H) = \mathbb{E}_{\mathbf{d} = \langle P, c \rangle \sim Q} \min_{h \in H} \mathbb{E}_{\mathbf{x} \sim P}\ell\left[h(\mathbf{x}), c(\mathbf{x})\right]. \tag{25}$$
The derivation of proposition (i) from proposition (ii) directly follows from the definition (25) by expanding on the first expectation operator over the domain distribution $Q$.

Reversely, we prove proposition (ii) from proposition (i) by contradiction. Suppose proposition (ii) is false. Then, there exists $d_i = \langle P_i, c_i \rangle \in D$ $(i \in [N])$ such that for all $h_k \in H$ $(k \in [K])$, $\mathbb{E}_{\mathbf{x} \sim P_i}\ell\left[h_k(\mathbf{x}), c_i(\mathbf{x})\right] > 0$ holds. We thus have
$$\begin{aligned} er(H) &= \mathbb{E}_{\mathbf{d} = \langle P, c \rangle \sim Q} \min_{h \in H} \mathbb{E}_{\mathbf{x} \sim P}\ell\left[h(\mathbf{x}), c(\mathbf{x})\right] \\ &\geq Q(d_i) \min_{h \in H} \mathbb{E}_{\mathbf{x} \sim P_i}\ell\left[h(\mathbf{x}), c_i(\mathbf{x})\right] \\ &> 0. \end{aligned} \tag{26}$$

This is in contradiction with proposition (i). Therefore, proposition (ii) must hold if proposition (i) is true. This completes the proof. $\qquad\square$

### C.2   Proof of Theorem 2

We start by revisiting the definition of PAC learnability (Valiant, 1984).

**Definition 4** (PAC learnability, single-domain case). *A target function class (concept class) $\mathcal{C}$ is said to be PAC-learnable if there exists an algorithm $\mathcal{A}$ and a polynomial function $\mathrm{poly}(\cdot, \cdot, \cdot, \cdot)$ such that for any $\epsilon > 0$ and $\delta > 0$, for all distributions $P$ on $\mathcal{X}$ and for any target function $c \in \mathcal{C}$, the following holds for any sample size $l \geq \mathrm{poly}\left(1/\epsilon, 1/\delta, \kappa, \mathrm{size}(c)\right)$:*

$$\mathbb{P}_{S \sim P^l}\left[\mathbb{E}_{\mathbf{x} \sim P}\ell\left[h_S(\mathbf{x}), c(\mathbf{x})\right] \leq \epsilon\right] \geq 1 - \delta, \tag{27}$$

*where $h_S \in \mathcal{H}$ is the hypothesis returned by algorithm $\mathcal{A}$ after receiving a labeled sample $S$, $\kappa$ is a number such that the computational cost of representing any element $x \in \mathcal{X}$ is at most $O(\kappa)$, and $\mathrm{size}(c)$ denotes the maximal cost of the computational representation of $c \in \mathcal{C}$.*

Since the goal of our setting is to learn *multiple* target functions simultaneouslly (correponding to all target functions in the domain set), we introduce a natural extension of Definition 4 to the multi-domain case. Given a domain set $D = \{\langle P_i, c_i \rangle\}_{i=1}^N$ with cardinality $N$ and confliting rank $R(D)$, we denote by $C := \{c \mid \langle P, c \rangle \in D\}$ its corresponding *target function set* that contains all target functions belonging to the domains in the domain set $D$, and denote by $\mathbb{C}$ a target function set class in which each element is a target function set. Following the formulation in Sec. 2.1, we allow the algorithm $\mathcal{A}$ to return a *hypothesis set* $H_S = \{h_i\}_{i=1}^K \in \mathcal{H}^K$ (rather than a single hypothesis as in the standard PAC learning model) given a labeled sample $S$ that is drawn according to the bilevel sampling process as in Sec. 2.1.

**Definition 5** (PAC learnability, multi-domain case). *A target function set class $\mathbb{C}$ is said to be PAC-learnable if there exists an algorithm $\mathcal{A}$ and a polynomial function $\mathrm{poly}(\cdot, \cdot, \cdot, \cdot)$ such that for any $\epsilon > 0$ and $\delta > 0$, for all distributions $Q$ on $D = \{\langle P_i, c_i \rangle\}_{i=1}^N$, for all distributions $P_1, \cdots, P_N$ on $\mathcal{X}$, and for any target function set $C \in \mathbb{C}$, the following holds for any sample size $mn \geq \mathrm{poly}(1/\epsilon, 1/\delta, \kappa, \mathrm{size}(C))$:*

$$\forall c \in C, \exists h \in H_S \text{ such that } \mathbb{P}_S\left[\mathbb{E}_{\mathbf{x} \sim P}\ell\left[h(\mathbf{x}), c(\mathbf{x})\right] \leq \epsilon\right] \geq 1 - \delta, \tag{28}$$

*where $H_S \in \mathcal{H}^K$ is the hypothesis set returned by algorithm $\mathcal{A}$ after receiving a labeled sample $S$, $\kappa$ is a number such that the computational cost of representing any element $x \in \mathcal{X}$ is at most $O(\kappa)$, and $\mathrm{size}(C)$ denotes the maximal cost of the computational representation of $C \in \mathbb{C}$.*

Before proving Theorem 2, we first prove a lemma:

**Lemma 1.** *If LEAF is PAC-learnable according to Definition 5, then there exists an algorithm $\mathcal{A}$ such that for any $\delta > 0$, the hypothesis set $H_S$ returned by $\mathcal{A}$ after receiving a labeled sample $S$ with infinite examples satisfies*

$$\mathbb{P}_S\left[er(H_S) = 0\right] \geq 1 - \delta. \tag{29}$$

*Proof.* Recall the definition of the expected error $er(H)$ (4) under the expected allocation function (7b):

$$er(H) = \mathbb{E}_{\mathbf{d} = \langle P, c \rangle \sim Q} \min_{h \in H} \mathbb{E}_{\mathbf{x} \sim P}\ell\left[h(\mathbf{x}), c(\mathbf{x})\right]. \tag{30}$$

Suppose that LEAF is PAC-learnable. Then, Equation (28) holds for the sample size $mn \geq \mathrm{poly}(1/\epsilon, 1/\delta, \kappa, \mathrm{size}(C))$. By setting $\epsilon \to 0$, we have that for any $\delta > 0$,

$$\forall c \in C, \exists h \in H_S \text{ such that } \mathbb{P}_S\left[\mathbb{E}_{\mathbf{x} \sim P}\ell\left[h(\mathbf{x}), c(\mathbf{x})\right] = 0\right] \geq 1 - \delta. \tag{31}$$

holds when the sample size is infinite. This equals to

$$\mathbb{P}_S\left[\min_{h \in H_S} \mathbb{E}_{\mathbf{x} \sim P}\ell\left[h(\mathbf{x}), c(\mathbf{x})\right] = 0\right] \geq 1 - \delta \tag{32}$$

for every $d = \langle P, c \rangle \in D$. By replacing $\delta$ with $\delta/N$ in Equation (32) and a union bound argument, we have that for any distribution $Q$ on $D$,

$$\mathbb{P}_S\left[\mathbb{E}_{\mathbf{d} = \langle P, c \rangle \sim Q} \min_{h \in H_S} \mathbb{E}_{\mathbf{x} \sim P}\ell\left[h(\mathbf{x}), c(\mathbf{x})\right] = 0\right] \geq 1 - \delta \tag{33}$$

holds. Combining Equations (30) (33) completes the proof. $\square$

Now we are ready to prove Theorem 2.

**Theorem 2** (PAC learnability). *For a domain set $D$ with conflict rank $R(D)$ and a model set with cardinality $K$, a necessary condition of the PAC learnability of LEAF is $K \geq R(D)$.*

*Proof.* Recall the definition of conflict rank (Definition 2):

$$R(D) = \max_{\mathcal{S} \subseteq \mathcal{X}, \lambda(\mathcal{S}) > 0} \min_{x \in \mathcal{S}} \left| \left\{ c(x) \mid \langle P, c \rangle \in D, x \in \mathrm{supp}(P) \right\} \right|, \quad (34)$$

where $\lambda(\mathcal{S})$ is the Lebesgue measure of $\mathcal{S}$. In what follows, we drop the dependency of $R(D)$ on the domain set $D$ for simplicity, and denote the conflict rank of the domain set by $R$. From Equation (34), we have that there exists a subset $\mathcal{S} \subseteq \mathcal{X}$ such that $\lambda(\mathcal{S}) > 0$, and there are $R$ distinct domains $d_{(1)} = \langle P_{(1)}, c_{(1)} \rangle, \cdots, d_{(R)} = \langle P_{(R)}, c_{(R)} \rangle$ in the domain set $D$ satisfying

$$c_{(1)}(x) \neq c_{(2)}(x) \neq \cdots \neq c_{(R)}(x) \quad (35)$$

for all $x \in \mathcal{S}$. We now prove the main result by contradiction. Suppose that $K < R$, by the drawer principle, we have that there exists $c^\dagger \in \left\{ c_{(1)}, c_{(2)}, \cdots, c_{(R)} \right\}$ such that

$$\forall h \in H, \ c^\dagger(x) \neq h(x) \quad (36)$$

holds for all $x \in \mathcal{S}$. We then have that

$$
\begin{aligned}
er(H) &= \mathbb{E}_{\mathbf{d} = \langle P, c \rangle \sim Q} \min_{h \in H} \mathbb{E}_{\mathbf{x} \sim P} \ell \left[ h(\mathbf{x}), c(\mathbf{x}) \right] \\
&\geq Q\left( d^\dagger \right) \min_{h \in H} \mathbb{E}_{\mathbf{x} \sim P^\dagger} \ell \left[ h(\mathbf{x}), c^\dagger(\mathbf{x}) \right] \\
&= Q\left( d^\dagger \right) \min_{h \in H} \int_{x \in \mathcal{X}} \ell \left[ h(x), c^\dagger(x) \right] \, \mathrm{d}P^\dagger(x) \\
&\geq Q\left( d^\dagger \right) \min_{h \in H} \int_{x \in \mathcal{S}} \ell \left[ h(x), c^\dagger(x) \right] \, \mathrm{d}P^\dagger(x) \\
&> 0
\end{aligned}
\quad (37)
$$

holds with probability 1. On the other hand, by Lemma 1 we have that if LEAF is PAC-learnable, then there exists an algorithm that outputs a hypothesis set $H_S$ satisfying $er(H_S) = 0$ with probability $1 - \delta$ for any $\delta > 0$, which is in contradiction with (37). Therefore, we must have $K \geq R$. $\quad\square$

### C.3 GENERALIZED THEOREM 3 AND PROOF

In this section, we state and prove a generalized version of Theorem 3 in the main text. In Sec. 2.1, we formulate LEAF by a bilevel sampling procedure: first, $m$ domains are i.i.d. drawn from a domain distribution $Q$ over the domain set $D$, resulting in $m$ batches; then, for each batch, $n$ inputs are i.i.d. drawn from the corresponding domain's input distribution and labeled by its target function. Here we relax the second step of this sampling process by allowing the data in each batch to be drawn from *other* domains apart from the sampled domain. This models a general and more realistic scenario where each data batch is not absolutely "clean" and may contain a small portion of "noisy" data from other domains.

Formally, we introduce a *mixing ratio* parameter $\alpha \in [0, 1)$ that captures the expected percentage of noisy examples: for each data pair in the $i$-th ($i \in [n]$) batch, with probability $1 - \alpha$ it is drawn from the sampled domain $\mathbf{d}^i = \langle P^i, c^i \rangle$, and with probability $\alpha$ it is drawn from other domains in the domain set, i.e., $D'^i \subset D \setminus \{\mathbf{d}^i\}$. This amounts to sampling from a mixture of input distributions $P_{\mathrm{mix}}^i = (1 - \alpha)P^i + \alpha \tilde{P}^i$, where $\tilde{P}^i$ is the input distribution of one or a mixture of multiple domains from $D \setminus \{\mathbf{d}^i\}$. Correspondingly, we also introduce a "mixed" target function denoted by $c_{\mathrm{mix}}^i = (1 - \alpha)c^i + \alpha \tilde{c}^i$, which indicates that the input will be labeled by $c^i$ if sampled from $P^i$, while labeled by $\tilde{c}^i$ if sampled from $\tilde{P}^i$.

Clearly, the mixing ratio $\alpha$ needs to be small for effective training, but we would also like to theoretically investigate how the generalization error of LEAF relates to $\alpha$.

Now we are ready to state Theorem 4, which is a generalization of Theorem 3 in the main text:

**Theorem 4** (Generalization error upper bound). *For any mixing ratio $\alpha \in [0,1)$ and any $\delta \in (0,1]$, the following inequality holds uniformly for all model sets $H \in \mathcal{H}^K$ with probability at least $1 - \delta$:*

$$er(H) \leq \frac{1}{1-\alpha}\,\widehat{er}(H) + \frac{1+\alpha}{1-\alpha}\left(\sqrt{\frac{\mathscr{D}(\bar{\mathcal{S}})\left(\ln 2m/\mathscr{D}(\bar{\mathcal{S}}) + 1\right) - \ln \delta/24}{m}} + \frac{1}{m}\right) \tag{38a}$$

$$+ \frac{1}{1-\alpha}\sum_{k=1}^{N}\left(\frac{m_k}{m}\sqrt{\frac{\mathscr{D}(\mathcal{S})\left(\ln 2m_k n/\mathscr{D}(\mathcal{S}) + 1\right) - \ln \delta/12N}{m_k n}} + \frac{1}{mn}\right) \tag{38b}$$

$$+ \frac{2}{1-\alpha}\left(\sqrt{\frac{\mathscr{D}(\mathcal{S})\left(\ln 2n/\mathscr{D}(\mathcal{S}) + 1\right) - \ln \delta/24m}{n}} + \frac{1}{n}\right), \tag{38c}$$

*where $\bar{\mathcal{S}} = \{\min_{h \in \mathcal{H}}\langle P, c\rangle \mapsto \mathbb{E}_{\mathbf{x} \sim P}\ell\left[h(\mathbf{x};\Theta), c(\mathbf{x})\right]\}$ is the function set of the domain-wise expected error, $\mathcal{S} = \{(x,y) \mapsto \ell\left[h(x;\theta), y\right]\}$ is the function set of the instance-wise error, $m_k = \sum_{i=1}^{m}\mathbb{1}\left(\mathbf{d}^i = d_k\right), k \in [N]$ is the count of the $k$-th domain in the domain set $D$ during the sampling process, and $\mathscr{D}(\cdot)$ is the Vapnik-Chervonenkis (VC) dimension (Vapnik & Chervonenkis, 1971) for some set of real functions.[2]*

**Remark 3.** It is straightforward to recover Theorem 3 in the main text using Theorem 4 by setting the mixing ratio $\alpha = 0$.

### C.3.1 TECHNICAL LEMMAS AND INTERMEDIATE RESULTS

Before giving the proof of Theorem 4 (generalized Theorem 3), we first introduce several technical lemmas.

**Lemma 2.** *Let $\{E_i\}_{i=1}^{n}$ be a set of events satisfying $\mathbb{P}(E_i) \geq 1 - \delta_i$, with some $\delta_i \geq 0, i = 1, \cdots, n$. Then, $\mathbb{P}\left(\bigcap_{i=1}^{n} E_i\right) \geq 1 - \sum_{i=1}^{n}\delta_i$.*

*Proof.* Straightforward using the union bound. $\qquad\square$

The next lemma is a well-known result in statistical learning theory Vapnik (1999), and is a standard tool for upper-bounding the generalization error of empirical and expected data distributions:

**Lemma 3** (Single-task generalization error upper bound (Vapnik, 1999)). *Let $\mathcal{Z}$ be a space, and let $\Lambda$ be a parameter space. Let $A \leq \mathcal{Q}(z;\alpha) \leq B, \alpha \in \Lambda$ be a measurable and bounded real function set, of which the VC-dimension is $\mathscr{D}(\mathcal{Q})$. Let $\{\mathbf{z}_i\}_{i=1}^{n}$ be the data set sampled i.i.d. from a distribution $P$ on $\mathcal{Z}$ with size $n$. Then, for any $\delta \in (0,1]$, the following inequality holds with probability at least $1 - \delta$:*

$$\left|\mathbb{E}_{\mathbf{z} \sim P}\mathcal{Q}(\mathbf{z};\alpha) - \frac{1}{n}\sum_{i=1}^{n}\mathcal{Q}\left(\mathbf{z}_i;\alpha\right)\right| \leq (B - A)\sqrt{\epsilon(n)} + \frac{1}{n}, \tag{39}$$

*where*

$$\epsilon(n) = \frac{\mathscr{D}(\mathcal{Q})\left(\ln 2n/\mathscr{D}(\mathcal{Q}) + 1\right) - \ln \delta/4}{n}. \tag{40}$$

Given the technical lemmas, we now present some intermediate results that will be necessary in the proof of Theorem 4. We define the following shorthand:

$$h_{i,\text{clean}}^* \in \underset{h \in H}{\arg\min}\,\mathbb{E}_{\mathbf{x} \sim P^i}\ell\left[h(\mathbf{x}), c^i(\mathbf{x})\right], \tag{41a}$$

$$\hat{h}_i^* \in \underset{h \in H}{\arg\min}\sum_{j=1}^{n}\ell\left[h\left(\mathbf{x}^{ij}\right), \mathbf{y}^{ij}\right], \tag{41b}$$

---

[2]The standard VC-dimension is defined on sets of indicator functions and thereby only applies to the binary classification setting. Here we adopt a generalized notion of VC-dimension, which is defined on real function sets and thus applies to general loss functions, while having no impact on the forms of the generalization error bounds. See Chapter 3 of (Vapnik, 1999) for more details and discussion.

where $i \in [m]$ denotes the $i$-th batch of LEAF, and $H$ is the model set. Note that here the $\arg\max$ operator is defined on the *model set $H$* rather than the hypothesis space $\mathcal{H}$. For each batch with index $i$, we also define a clean data batch $Z_{\text{clean}}^i = \{(\mathbf{x}_{\text{clean}}^{ij}, \mathbf{y}_{\text{clean}}^{ij})\}_{j=1}^n$ that is i.i.d. drawn from $\mathbf{d}^i$, corresponding to the case with mixing ratio $\alpha = 0$. Recall that we use superscripts to denote the sampling index, e.g., $\mathbf{d}^i = \langle P^i, c^i \rangle$ denotes the $i$-th domain sample, which can be any domain in the domain set $D$.

We first prove a lemma that upper-bounds the generalization error induced by the discrepancy between the empirical and expected domain distributions:

**Lemma 4** (Domain estimation error). *Let $Q$ be a distribution over the domain set $D$, and let $\mathbf{d}^1 = \langle P^1, c^1 \rangle, \cdots, \mathbf{d}^m = \langle P^m, c^m \rangle$ be domain examples i.i.d. drawn from $Q$. Then, for any $\delta \in (0, 1]$, the following inequality holds uniformly for all model sets $H \in \mathcal{H}^K$ with probability at least $1 - \delta$:*

$$\left| \frac{1}{m} \sum_{i=1}^m \min_{h \in H} \mathbb{E}_{\mathbf{x} \sim P^i} \ell\left[h(\mathbf{x}), c^i(\mathbf{x})\right] - \mathbb{E}_{d_i \sim Q} \min_{h \in H} \mathbb{E}_{\mathbf{x} \sim P_i} \ell\left[h(\mathbf{x}), c_i(\mathbf{x})\right] \right| \tag{42}$$

$$\leq \sqrt{\frac{\mathscr{D}(\bar{\mathcal{S}}) \ln\left(2m/\mathscr{D}(\bar{\mathcal{S}}) + 1\right) - \ln \delta/4}{m}} + \frac{1}{m}.$$

*where $\bar{\mathcal{S}} = \{\langle P, c \rangle \mapsto \min_{h \in H} \mathbb{E}_{\mathbf{x} \sim P} \ell\left[h(\mathbf{x}; \Theta), c(\mathbf{x})\right]\}$ is the function set of the domain-wise expected error, and $\mathscr{D}(\cdot)$ is the VC-dimension of some set of real functions.*

*Proof.* We define a "domain-averaged" loss function over the model set $H$ and the domain $d = \langle P, c \rangle$:

$$\bar{\ell}(H, d) := \min_{h \in H} \mathbb{E}_{\mathbf{x} \sim P} \ell\left[h(\mathbf{x}), c(\mathbf{x})\right].$$

Then, we have

$$\frac{1}{m} \sum_{i=1}^m \min_{h \in H} \mathbb{E}_{\mathbf{x} \sim P^i} \ell\left[h(\mathbf{x}), c^i(\mathbf{x})\right] - \mathbb{E}_{d_i \sim Q} \min_{h \in H} \mathbb{E}_{\mathbf{x} \sim P_i} \ell\left[h(\mathbf{x}), c_i(\mathbf{x})\right]$$

$$= \frac{1}{m} \sum_{i=1}^m \bar{\ell}\left(H, \mathbf{d}^i\right) - \mathbb{E}_{d_i \sim Q} \bar{\ell}(H, d_i),$$

Finally, replacing the function set $\mathcal{Q}$ in Lemma 3 with $\bar{\mathcal{S}} = \{\langle P, c \rangle \mapsto \min_{h \in H} \mathbb{E}_{\mathbf{x} \sim P} \ell\left[h(\mathbf{x}; \Theta), c(\mathbf{x})\right]\}$ completes the proof. $\qquad\square$

We then prove a lemma that upper-bounds the empirical error induced by the discrepancy between the empirical allocation function (7a) and the expected allocation function (7b):

**Lemma 5** (Allocation estimation error). *Let $\{(\mathbf{x}^{ij}, \mathbf{y}^{ij})\}_{j=1}^n$ be the data batch drawn in the $i$-th $(i \in [m])$ batch of LEAF with mixing ratio $\alpha$. Then, for any $\alpha \in [0, 1)$ and any $\delta \in (0, 1]$, the following inequality holds uniformly for all model sets $H \in \mathcal{H}^K$ with probability at least $1 - \delta$:*

$$\left| \frac{1}{m} \sum_{i=1}^m \frac{1}{n} \sum_{j=1}^n \ell\left[h_{i,\text{clean}}^*\left(\mathbf{x}_{\text{clean}}^{ij}\right), \mathbf{y}_{\text{clean}}^{ij}\right] - \frac{1}{m} \sum_{i=1}^m \frac{1}{n} \sum_{j=1}^n \ell\left[\hat{h}_i^*\left(\mathbf{x}^{ij}\right), \mathbf{y}^{ij}\right] \right|$$

$$\leq \alpha \cdot er(H) + \alpha \cdot \sqrt{\frac{\mathscr{D}(\bar{\mathcal{S}}) \ln\left(2m/\mathscr{D}(\bar{\mathcal{S}}) + 1\right) - \ln \delta/12}{m}} + \frac{\alpha}{m} \tag{43}$$

$$+ 2\sqrt{\frac{\mathscr{D}(\mathcal{S})(\ln 2n/\mathscr{D}(\mathcal{S}) + 1) - \ln \delta/12m}{n}} + \frac{2}{n},$$

*where $\bar{\mathcal{S}} = \{\langle P, c \rangle \mapsto \min_{h \in H} \mathbb{E}_{\mathbf{x} \sim P} \ell\left[h(\mathbf{x}; \Theta), c(\mathbf{x})\right]\}$ is the function set of the domain-wise expected error, $\mathcal{S} = \{(x, y) \mapsto \ell\left[h(x; \theta), y\right]\}$ is the function set of the sample-wise error, and $\mathscr{D}(\cdot)$ is the VC-dimension of some set of real functions.*

*Proof.* We have the following decomposition:

$$\frac{1}{m}\sum_{i=1}^{m}\frac{1}{n}\sum_{j=1}^{n}\ell\left[h_{i,\text{clean}}^{*}\left(\mathbf{x}_{\text{clean}}^{ij}\right),\mathbf{y}_{\text{clean}}^{ij}\right]-\frac{1}{m}\sum_{i=1}^{m}\frac{1}{n}\sum_{j=1}^{n}\ell\left[\hat{h}_{i}^{*}\left(\mathbf{x}^{ij}\right),\mathbf{y}^{ij}\right]$$

$$=\frac{1}{m}\sum_{i=1}^{m}\left\{\frac{1}{n}\sum_{j=1}^{n}\ell\left[h_{i,\text{clean}}^{*}\left(\mathbf{x}_{\text{clean}}^{ij}\right),\mathbf{y}_{\text{clean}}^{ij}\right]-\mathbb{E}_{\mathbf{x}\sim P^{i}}\ell\left[h_{i,\text{clean}}^{*}(\mathbf{x}),c^{i}(\mathbf{x})\right]\right\}$$

$$+\frac{1}{m}\sum_{i=1}^{m}\left\{\mathbb{E}_{\mathbf{x}\sim P^{i}}\ell\left[h_{i,\text{clean}}^{*}(\mathbf{x}),c^{i}(\mathbf{x})\right]-\mathbb{E}_{\mathbf{x}\sim P_{\text{mix}}^{i}}\ell\left[\hat{h}_{i}^{*}(\mathbf{x}),c_{\text{mix}}^{i}(\mathbf{x})\right]\right\} \qquad (44)$$

$$+\frac{1}{m}\sum_{i=1}^{m}\left\{\mathbb{E}_{\mathbf{x}\sim P_{\text{mix}}^{i}}\ell\left[\hat{h}_{i}^{*}(\mathbf{x}),c_{\text{mix}}^{i}(\mathbf{x})\right]-\frac{1}{n}\sum_{j=1}^{n}\ell\left[\hat{h}_{i}^{*}\left(\mathbf{x}^{ij}\right),\mathbf{y}^{ij}\right]\right\},$$

in which the original difference is decomposed into three terms in the RHS.

We can use the following technique to bound both the first term and the last term: by substituting $\mathcal{Q}$ in Lemma 3 with $\mathcal{S}=\{(x,y)\mapsto\ell\left[h(x;\theta),y\right]\}$ and replacing $\delta$ with $\delta/3m$, every term in the summation operator can be upper-bounded by $\sqrt{\dfrac{\mathscr{D}(\mathcal{S})\ln\left(2n/\mathscr{D}(\mathcal{S})+1\right)-\ln\delta/12m}{n}}+\dfrac{1}{n}$ with probability at least $1-\delta/3m$; we then combine the terms in the summation operator using Lemma 2, showing that both the first term and the last term in the RHS of (44) can be bounded by $\sqrt{\dfrac{\mathscr{D}(\mathcal{S})\ln\left(2n/\mathscr{D}(\mathcal{S})+1\right)-\ln\delta/12m}{n}}+\dfrac{1}{n}$ with probability at least $1-\delta/3$.

There remains the middle term in the RHS of (44), for which we have

$$\frac{1}{m}\sum_{i=1}^{m}\left\{\mathbb{E}_{\mathbf{x}\sim P^{i}}\ell\left[h_{i,\text{clean}}^{*}(\mathbf{x}),c^{i}(\mathbf{x})\right]-\mathbb{E}_{\mathbf{x}\sim P_{\text{mix}}^{i}}\ell\left[\hat{h}_{i}^{*}(\mathbf{x}),c_{\text{mix}}^{i}(\mathbf{x})\right]\right\}$$

$$=\frac{1-\alpha}{m}\sum_{i=1}^{m}\left\{\mathbb{E}_{\mathbf{x}\sim P^{i}}\ell\left[h_{i,\text{clean}}^{*}(\mathbf{x}),c^{i}(\mathbf{x})\right]-\mathbb{E}_{\mathbf{x}\sim P^{i}}\ell\left[\hat{h}_{i}^{*}(\mathbf{x}),c^{i}(\mathbf{x})\right]\right\} \qquad (45)$$

$$+\frac{\alpha}{m}\sum_{i=1}^{m}\left\{\mathbb{E}_{\mathbf{x}\sim P^{i}}\ell\left[h_{i,\text{clean}}^{*}(\mathbf{x}),c^{i}(\mathbf{x})\right]-\mathbb{E}_{\mathbf{x}\sim\tilde{P}^{i}}\ell\left[\hat{h}_{i}^{*}(\mathbf{x}),\tilde{c}^{i}(\mathbf{x})\right]\right\}$$

By the definitions in (41a) and (41b), the first term in the RHS of (45) is non-positive and thus can be upper-bounded by zero. For the other term, we have

$$\frac{\alpha}{m}\sum_{i=1}^{m}\left\{\mathbb{E}_{\mathbf{x}\sim P^{i}}\ell\left[h_{i,\text{clean}}^{*}(\mathbf{x}),c^{i}(\mathbf{x})\right]-\mathbb{E}_{\mathbf{x}\sim\tilde{P}^{i}}\ell\left[\hat{h}_{i}^{*}(\mathbf{x}),\tilde{c}^{i}(\mathbf{x})\right]\right\}$$

$$\leq\frac{\alpha}{m}\sum_{i=1}^{m}\mathbb{E}_{\mathbf{x}\sim P^{i}}\ell\left[h_{i,\text{clean}}^{*}(\mathbf{x}),c^{i}(\mathbf{x})\right]$$

$$=\alpha\cdot er(H)+\alpha\left\{\frac{1}{m}\sum_{i=1}^{m}\mathbb{E}_{\mathbf{x}\sim P^{i}}\ell\left[h_{i,\text{clean}}^{*}(\mathbf{x}),c^{i}(\mathbf{x})\right]-\mathbb{E}_{d_{i}\sim Q}\min_{h\in H}\mathbb{E}_{\mathbf{x}\sim P_{i}}\ell\left[h(\mathbf{x}),c_{i}(\mathbf{x})\right]\right\}$$

where the inequality is due to the nonnegativity of the loss function $\ell$. By replacing $\delta$ with $\delta/3$ in Lemma 4, the second term in RHS of the last line can be upper-bounded by $\alpha\cdot\sqrt{\dfrac{\mathscr{D}(\bar{\mathcal{S}})\ln\left(2m/\mathscr{D}(\bar{\mathcal{S}})+1\right)-\ln\delta/12}{m}}+\dfrac{\alpha}{m}$ with probability at least $1-\delta/3$.

Finally, combining the above three bounds corresponding to the terms in the RHS of (44) using Lemma 2 completes the proof. $\square$

### C.3.2 PROOF OF THEOREM 4 (GENERALIZED THEOREM 3)

Now we are ready to give the proof of Theorem 4.

*Proof.* Combining the empirical and expected objectives (3) (4) with the allocation functions (7a) (7b), we have

$$er(H) - \widehat{er}(H) = \mathbb{E}_{d_i \sim Q} \min_{h \in H} \mathbb{E}_{\mathbf{x} \sim P_i} \ell\left[h(\mathbf{x}), c_i(\mathbf{x})\right] - \frac{1}{m} \sum_{i=1}^{m} \min_{h \in H} \frac{1}{n} \sum_{j=1}^{n} \ell\left[h\left(\mathbf{x}^{ij}\right), \mathbf{y}^{ij}\right], \quad (46)$$

where for every $i \in [m]$, the data $\{(\mathbf{x}^{ij}, \mathbf{y}^{ij})\}_{j=1}^{n}$ is sampled from $P_{\text{mix}}^{i} = (1-\alpha)P^i + \alpha \tilde{P}^i$ and labeled by $c_{\text{mix}}^{i} = (1-\alpha)c^i + \alpha \tilde{c}^i$ with mixing ratio $\alpha \in [0, 1)$.

We then have the following decomposition:

$$er(H) - \widehat{er}(H) = \left\{ \mathbb{E}_{d_i \sim Q} \min_{h \in H} \mathbb{E}_{\mathbf{x} \sim P_i} \ell\left[h(\mathbf{x}), c_i(\mathbf{x})\right] - \frac{1}{m} \sum_{i=1}^{m} \min_{h \in H} \mathbb{E}_{\mathbf{x} \sim P^i} \ell\left[h(\mathbf{x}), c^i(\mathbf{x})\right] \right\}$$

$$+ \left\{ \frac{1}{m} \sum_{i=1}^{m} \min_{h \in H} \mathbb{E}_{\mathbf{x} \sim P^i} \ell\left[h(\mathbf{x}), c^i(\mathbf{x})\right] - \frac{1}{m} \sum_{i=1}^{m} \frac{1}{n} \sum_{j=1}^{n} \ell\left[h_{i,\text{clean}}^{*}\left(\mathbf{x}_{\text{clean}}^{ij}\right), \mathbf{y}_{\text{clean}}^{ij}\right] \right\}$$

$$+ \left\{ \frac{1}{m} \sum_{i=1}^{m} \frac{1}{n} \sum_{j=1}^{n} \ell\left[h_{i,\text{clean}}^{*}\left(\mathbf{x}_{\text{clean}}^{ij}\right), \mathbf{y}_{\text{clean}}^{ij}\right] - \frac{1}{m} \sum_{i=1}^{m} \min_{h \in H} \frac{1}{n} \sum_{j=1}^{n} \ell\left[h\left(\mathbf{x}^{ij}\right), \mathbf{y}^{ij}\right] \right\}.$$
$$(47)$$

Applying the definitions in (41a) and (41b), we rewrite the equation above:

$$er(H) - \widehat{er}(H) = \left\{ \mathbb{E}_{d_i \sim Q} \min_{h \in H} \mathbb{E}_{\mathbf{x} \sim P_i} \ell\left[h(\mathbf{x}), c_i(\mathbf{x})\right] - \frac{1}{m} \sum_{i=1}^{m} \min_{h \in H} \mathbb{E}_{\mathbf{x} \sim P^i} \ell\left[h(\mathbf{x}), c^i(\mathbf{x})\right] \right\}$$

$$+ \left\{ \frac{1}{m} \sum_{i=1}^{m} \mathbb{E}_{\mathbf{x} \sim P^i} \ell\left[h_{i,\text{clean}}^{*}(\mathbf{x}), c^i(\mathbf{x})\right] - \frac{1}{m} \sum_{i=1}^{m} \frac{1}{n} \sum_{j=1}^{n} \ell\left[h_{i,\text{clean}}^{*}\left(\mathbf{x}_{\text{clean}}^{ij}\right), \mathbf{y}_{\text{clean}}^{ij}\right] \right\}$$

$$+ \left\{ \frac{1}{m} \sum_{i=1}^{m} \frac{1}{n} \sum_{j=1}^{n} \ell\left[h_{i,\text{clean}}^{*}\left(\mathbf{x}_{\text{clean}}^{ij}\right), \mathbf{y}_{\text{clean}}^{ij}\right] - \frac{1}{m} \sum_{i=1}^{m} \frac{1}{n} \sum_{j=1}^{n} \ell\left[\hat{h}_i^{*}\left(\mathbf{x}^{ij}\right), \mathbf{y}^{ij}\right] \right\},$$
$$(48)$$

in which the generalization error of LEAF is decomposed into three terms. These terms represent different aspects that impact the generalization error:

- The first term represents the discrepancy between the expected and empirical domain distributions.

- The second term represents the discrepancy between the expected and empirical data distributions in each batch.

- The last term represents the discrepancy between the expected and empirical allocation functions. Note that when the mixing ratio $\alpha > 0$, this term also accounts for the discrepancy induced by the noise in the data batches.

In the sequel, we bound these terms separately to derive the final generalization error upper bound:

**Bounding the first term.** By replacing $\delta$ with $\delta/6$ in Lemma 4, we have that

$$\mathbb{E}_{d_i \sim Q} \min_{h \in H} \mathbb{E}_{\mathbf{x} \sim P_i} \ell\left[h(\mathbf{x}), c_i(\mathbf{x})\right] - \frac{1}{m} \sum_{i=1}^{m} \min_{h \in H} \mathbb{E}_{\mathbf{x} \sim P^i} \ell\left[h(\mathbf{x}), c^i(\mathbf{x})\right]$$

$$\leq \sqrt{\frac{\mathscr{D}(\bar{\mathcal{S}}) \ln\left(2m/\mathscr{D}(\bar{\mathcal{S}}) + 1\right) - \ln \delta/24}{m}} + \frac{1}{m} \tag{49}$$

holds uniformly for all $H \in \mathcal{H}^K$ with probability at least $1 - \delta/6$.

**Bounding the second term.** By rearranging the terms and use $m_k = \sum_{i=1}^{m} \mathbb{1}\left(c^i = c_k\right)$, $i \in [m]$, $k \in [N]$, we have

$$\frac{1}{m} \sum_{i=1}^{m} \mathbb{E}_{\mathbf{x} \sim P^i} \ell\left[h_{i,\text{clean}}^*(\mathbf{x}), c^i(\mathbf{x})\right] - \frac{1}{m} \sum_{i=1}^{m} \frac{1}{n} \sum_{j=1}^{n} \ell\left[h_{i,\text{clean}}^*\left(\mathbf{x}_{\text{clean}}^{ij}\right), \mathbf{y}_{\text{clean}}^{ij}\right]$$

$$= \frac{1}{m} \sum_{i=1}^{m} \left\{ \mathbb{E}_{\mathbf{x} \sim P^i} \ell\left[h_{i,\text{clean}}^*(\mathbf{x}), c^i(\mathbf{x})\right] - \frac{1}{n} \sum_{j=1}^{n} \ell\left[h_{i,\text{clean}}^*\left(\mathbf{x}_{\text{clean}}^{ij}\right), \mathbf{y}_{\text{clean}}^{ij}\right] \right\}$$

$$= \frac{1}{m} \sum_{k=1}^{N} \left\{ m_k \cdot \mathbb{E}_{\mathbf{x} \sim P_k} \ell\left[h_{k,\text{clean}}^*(\mathbf{x}), c_k(\mathbf{x})\right] - \frac{1}{n} \sum_{j=1}^{nm_k} \ell\left[h_{k,\text{clean}}^*\left(\mathbf{x}_{k,\text{clean}}^{j}, \mathbf{y}_{k,\text{clean}}^{j}\right)\right] \right\}$$

$$= \frac{1}{m} \sum_{k=1}^{N} m_k \left\{ \mathbb{E}_{\mathbf{x} \sim P_k} \ell\left[h_{k,\text{clean}}^*(\mathbf{x}), c_k(\mathbf{x})\right] - \frac{1}{nm_k} \sum_{j=1}^{nm_k} \ell\left[h_{k,\text{clean}}^*\left(\mathbf{x}_{k,\text{clean}}^{j}, \mathbf{y}_{k,\text{clean}}^{j}\right)\right] \right\}.$$

In the above, we rearrange the total $m$ data batches according to the domains they belong to. With a little abuse of notation, in the third and fourth lines we use $h_k^*$ to denote the hypothesis that yields the smallest expected error in the $k$-th domain in the domain set $D$, i.e., $h_k^* = \arg\min_{h \in H} \mathbb{E}_{\mathbf{x} \sim P_k} \ell\left[h(\mathbf{x}), c_k(\mathbf{x})\right]$, $k \in [N]$. The above transformation aggregates the domain examples so that the examples from the same domains that emerge multiple times can be accumulated and jointly considered, which leads to a sharper and more realistic error bound. In this way, we use $\{(\mathbf{x}_{k,\text{clean}}^{j}, \mathbf{y}_{k,\text{clean}}^{j})\}_{j=1}^{nm_k}$, $k \in [N]$ to denote the aggregated data that belongs to the $k$-th domain in $D$. By replacing $\delta$ with $\delta/3N$ in Lemma 3, for every $k \in [N]$ and $\delta \in (0, 1]$ we have that

$$\mathbb{E}_{x \sim P_k} \ell\left[h_k^*(x), c_k(x)\right] - \frac{1}{nm_k} \sum_{j=1}^{nm_k} \ell\left[h_k^*\left(x_{kj}, y_{kj}\right)\right]$$

$$\leq \sqrt{\frac{\mathscr{D}(\mathcal{S})\left(\ln 2m_k n / \mathscr{D}(\mathcal{S}) + 1\right) - \ln \delta/12N}{m_k n}} + \frac{1}{m_k n}$$

holds uniformly for all $H \in \mathcal{H}^K$ with probability at least $1 - \delta/3N$. By Lemma 2, we have that

$$\frac{1}{m} \sum_{i=1}^{m} \mathbb{E}_{\mathbf{x} \sim P^i} \ell\left[h_{i,\text{clean}}^*(\mathbf{x}), c^i(\mathbf{x})\right] - \frac{1}{m} \sum_{i=1}^{m} \frac{1}{n} \sum_{j=1}^{n} \ell\left[h_{i,\text{clean}}^*\left(\mathbf{x}_{\text{clean}}^{ij}\right), \mathbf{y}_{\text{clean}}^{ij}\right]$$

$$\leq \sum_{k=1}^{N} \left(\frac{m_k}{m} \sqrt{\frac{\mathscr{D}(\mathcal{S})\left(\ln 2m_k n / \mathscr{D}(\mathcal{S}) + 1\right) - \ln \delta/12N}{m_k n}} + \frac{1}{mn}\right) \tag{50}$$

holds uniformly for all $H \in \mathcal{H}^K$ with probability at least $1 - \delta/3$.

**Bounding the last term.** By replacing $\delta$ with $\delta/2$ in Lemma 5, we have that

$$\frac{1}{m} \sum_{i=1}^{m} \frac{1}{n} \sum_{j=1}^{n} \ell\left[h_{i,\text{clean}}^*\left(\mathbf{x}_{\text{clean}}^{ij}\right), \mathbf{y}_{\text{clean}}^{ij}\right] - \frac{1}{m} \sum_{i=1}^{m} \frac{1}{n} \sum_{j=1}^{n} \ell\left[\hat{h}_i^*\left(\mathbf{x}^{ij}\right), \mathbf{y}^{ij}\right]$$

$$\leq \alpha \cdot er(H) + \alpha \cdot \sqrt{\frac{\mathscr{D}(\bar{\mathcal{S}}) \ln\left(2m / \mathscr{D}(\bar{\mathcal{S}}) + 1\right) - \ln \delta/24}{m}} + \frac{\alpha}{m} \tag{51}$$

$$+ 2\sqrt{\frac{\mathscr{D}(\mathcal{S})\left(\ln 2n / \mathscr{D}(\mathcal{S}) + 1\right) - \ln \delta/24m}{n}} + \frac{2}{n}$$

holds uniformly for all $H \in \mathcal{H}^K$ with probability at least $1 - \delta/2$.

Finally, combining the above three bounds (49) (50) (51) using Lemma 2 gives that

$$
\begin{aligned}
er(H) - \widehat{er}(H) \leq & \sqrt{\frac{\mathscr{D}(\bar{\mathcal{S}}) \ln\left(2m/\mathscr{D}(\bar{\mathcal{S}}) + 1\right) - \ln \delta/24}{m}} + \frac{1}{m} \\
& + \sum_{k=1}^{N} \left( \frac{m_k}{m} \sqrt{\frac{\mathscr{D}(\mathcal{S}) \left(\ln 2m_k n/\mathscr{D}(\mathcal{S}) + 1\right) - \ln \delta/12N}{m_k n}} + \frac{1}{mn} \right) \\
& + \alpha \cdot er(H) + \alpha \cdot \sqrt{\frac{\mathscr{D}(\bar{\mathcal{S}}) \ln\left(2m/\mathscr{D}(\bar{\mathcal{S}}) + 1\right) - \ln \delta/24}{m}} + \frac{\alpha}{m} \\
& + 2\sqrt{\frac{\mathscr{D}(\mathcal{S})(\ln 2n/\mathscr{D}(\mathcal{S}) + 1) - \ln \delta/24m}{n}} + \frac{2}{n}
\end{aligned}
\tag{52}
$$

holds uniformly for all $H \in \mathcal{H}^K$ with probability at least $1 - \delta$. A simple transformation of the above inequality completes the proof. □

## D    EXPERIMENTAL DETAILS

In this section, we provide additional details on our experiments. All of our experiments are conducted based on PyTorch (Paszke et al., 2019) using a NVIDIA 2080ti GPU. The datasets we use can be downloaded from public data sources, and does not contain personally identifiable information or offensive content.

### D.1    REGRESSION

We consider a regression scenario in which the examples are randomly sampled from three heterogeneous functions in absolute, sinusoidal and logarithmic function families:

$$
\begin{aligned}
y &= 2\,|x| - 2 + \delta_{\text{noise}}, \ -2 \leq x \leq 2 \\
y &= 2 \sin\left(3x + \frac{\pi}{2}\right) + \delta_{\text{noise}}, \ -2 \leq x \leq 2 \\
y &= \frac{3}{2} \log\left(-x + \frac{5}{2}\right) - 1 + \delta_{\text{noise}}, \ -2 \leq x \leq 2,
\end{aligned}
\tag{53}
$$

where $\delta_{\text{noise}} \sim \mathcal{N}(0, 0.01^2)$ is the additive Gaussian noise. We consider each function as a domain, and set the parameters of LEAF as $m = 250, n = 2$, i.e., a total number of 500 examples are collected in 250 batches, each with two examples. In each batch, we randomly select a function and uniformly sample the examples from the selected function.

**Model and optimizer.** We use a model set with three neural network regressors with the same architecture. Each network consists of five fully-connected layers, with 32 neurons in each layer and ReLU nonlinearities between the layers. We use a standard SGD optimizer with $learning\ rate = 0.05$, $momentum = 0.9$, and $\ell_2\ weight\ decay = 10^{-4}$ to train these models.

#### D.1.1    BASELINE DETAILS

**MAML.** We use a standard PyTorch implememtation of MAML from GitHub[3]. We adopt the same network architecture for MAML as ours, and use the following hyperparameters:

$Shot = 2, Evalation = 100, Outer\ step\ size = 0.05, Inner\ step\ size = 0.015,$
$Inner\ grad\ steps = 2, Eval\ grad\ steps = 5, Eval\ iters = 10, Iterations = 20000.$

In each batch, we use two support examples and another two query examples for MAML to fine-tune its model. Note that this amounts to a batch size of four, which is larger than LEAF (with only two examples in each batch).

**Modular meta-learning.** We use the official PyTorch implementation of modular meta-learning from GitHub[4]. Since modular meta-learning uses more neural network modules than LEAF, we apply a smaller network for each module, with the following hyperparameters:

---

[3] https://github.com/dragen1860/MAML-Pytorch (MIT license)
[4] https://github.com/FerranAlet/modular-metalearning (MIT license)

$Shot = 2, Support = 2, Network = Linear \ 1 - 16 - 16 - 1, Num\_modules = 5,$
$Composer = Sum, meta\_lr = 0.003, Steps = 3000.$

Like MAML, in each batch we use two support examples and another two query examples for modular meta-learning for fine-tuning, resulting in a batch size of four.

**Sparse MoE.** We implement this baseline following the design in (Shazeer et al., 2017), where the output of the MoE model can be written as

$$y = \sum_{i=1}^{K} G(x)_i E_i(x), \tag{54}$$

where $G(x)$ and $E_i(x)$ are the output of the gating network and the output of the $i$-th expert network for the input $x$, respectively. We use the same number of experts as the number of low-level models in LEAF, and use the same network architecture as in LEAF for each expert. In the original work (Shazeer et al., 2017), the MoE model is used in language modeling tasks with no explict conflict in the training data, thus the gating network can modulate the outputs of experts only using the input $x$. However, when dealing with conflicting data, only having access to the input is not sufficient for gating since different examples may have distinct outputs for the same input. Therefore, we augment the input of the gating network with the ground-truth output $y$, resulting in

$$y = \sum_{i=1}^{K} G(x, y)_i E_i(x). \tag{55}$$

In (Shazeer et al., 2017), the authors choose a noisy top-$k$ gating strategy to obtain a sparse gating function. We follow this design and further set $k = 1$ as in LEAF, since we only want one model (expert) to be activated given an example. Concretely, the output of the gating network is given by

$$G(x, y) = \text{Softmax}(\text{KeepTop1}(H(x, y))), \tag{56}$$

where

$$H(x, y)_i = ([x, y] \cdot W_g)_i + \delta \cdot \log\left(1 + \exp\left([x, y] \cdot W_{\text{noise}})_i\right)\right),$$
$$\text{KeepTop1}(v)_i = \begin{cases} v_i, & \text{if } v_i \text{ is the largest element of } v. \\ -\infty, & \text{otherwise.} \end{cases}, \tag{57}$$

in which $\delta \sim \mathcal{N}(0, 1)$ is the noise. As shown in the above, the gating network involves two trainable weight matrices, $W_g$ and $W_{\text{noise}}$, which is trained end-to-end along with the training of all experts. This differs from LEAF that does not involve additional trainable parameters in the allocation function. During testing, we show the outputs of all experts.

**LEAF-EM.** We implement this variant in the same way as LEAF, expect that the allocation function is stochastic with the sampling strategy as in Sec. A.3.1. This sampling process involve an additional hyperparameter $\eta$ corresponding to the $2\sigma^2$ parameter in Equation (22); we set $\eta = 0.1$ in our experiments.

## D.2 CLASSIFICATION

As stated in the main text, we consider three classification settings that involve paralleled, hierarchical, and opposite input-output relations across the examples. In the sequel, we provide the experimental details on these settings separately.

**Paralleled relations.** For this setting, we use two datasets, namely *Colored MNIST* and *Fashion Product Images*.

*Colored MNIST* is an extended version of the classical digit recognition dataset MNIST. For each gray-scale digit image in the MNIST training set, we randomly color it using a pre-defined color pool consisting of eight colors: {*red*, *blue*, *yellow*, *green*, *pink*, *cyan*, *white*, *purple*}, and assign each colored digit with an additional color label. This results in two domains in the colored dataset with 60000 training images: *Digit* and *Color*, as shown in Figure 5. In each training batch, we randomly select one domain and inherit the ground-truth labels in the selected domain for all examples. We set the sampling parameters of LEAF to $m = 600000$, $n = 1$, corresponding to 10 training epochs. We

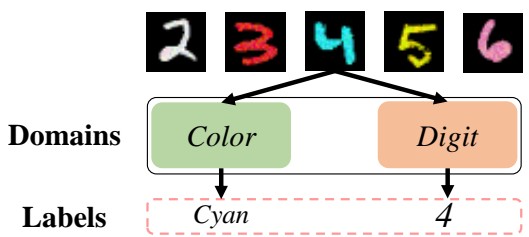

Figure 5: Paralleled relations: *Colored MNIST*.

use a model set with two neural network classifiers with the same architecture, consisting of three convolutional layers and one fully-connected layer with ReLU nonlinearities between the layers. We use the Adam optimizer (Kingma & Ba, 2015) with $learning\ rate = 0.002$ and $betas = (0.5, 0.999)$ without weight decay for training.

*Fashion Product Images* (Aggarwal, 2019) is a dataset for automatic attribute completion and question answering with multiple category labels. We choose three main parallel tasks with eight main labels from the original dataset, resulting in three domains namely *Color*, *Gender*, and *Category* containing labels {*black*, *white*, *blue*}, {*male*, *female*}, and {*apparel*, *footwear*, *accessories*}, respectively. There are totally 15000 training images. In each training batch, we randomly select one domain and inherit the ground-truth labels in the selected domain for all examples. We set the sampling parameters of LEAF to $m = 750000, n = 1$, corresponding to 50 training epochs. We use a model set consisting of three neural network classifiers with the same architecture. Each model has five convolutional layers and one fully-connected layer with ReLU nonlinearities between the layers. We use the Adam optimizer with $learning\ rate = 0.002$ and $betas = (0.5, 0.999)$ without weight decay for training.

**Hierarchical relations.** For this setting, we use the *CIFAR-100* dataset (Krizhevsky & Hinton, 2009), which is a widely-used benchmark for image recognition. It has a hierarchical structure with 20 superclasses and 100 classes, with 60000 images in total. Each superclass further contains 5 "fine" classes; for instance, the superclass *insect* contains *bee*, *beetle*, *butterfly*, *caterpillar*, and *cockroach*. In each training batch, we randomly select one domain and inherit the ground-truth labels in the selected domain for all examples. We set the sampling parameters of LEAF to $m = 1200000, n = 2$, corresponding to 40 training epochs. We use a model set with two neural network classifiers with the same architecture. For each model, we use a pre-trained DenseNet (Huang et al., 2017) backbone for feature extraction for LEAF and all baselines, and add two fully-connected layers after the backbone with ReLU nonlinearities. We use a standard SGD optimizer with $learning\ rate = 0.1$ and $momentum = 0.9$ with no weight decay to train the models.

**Opposite relations.** For this setting, we use a relabeled *Fashion Product Images* dataset to simulate the hidden context that depends on human preferences. Concretely, we re-label the *Fashion Product Images* dataset and construct a preference-dependent scenario with two domains: *Male* and *Female*. This simulates the scenario where the preference is based on the gender attribute. The re-labeling process is as follows: we randomly split the dataset into two domains according to the "gender" attribute: 50% of examples labeled as "male" and 50% examples labeled as "female" are assigned to the first domain with "male" re-labeled as "positive" and "female" re-labeled as "negative", and other examples are assigned to the second domain with "male" re-labeled as "negative" and "female" re-labeled as "positive". In each training batch, we randomly select one domain and use the "positive" or "negative" attribute of the examples as the ground-truth labels for classification. We set the sampling parameters of LEAF to $m = 37500, n = 20$, corresponding to 50 training epochs. We use a model set with two neural network classifiers with the same architecture. Each model consists of five convolutional layers and one fully-connected layer with ReLU nonlinearities as in the *Fashion Product Images* in the paralleled relation setting. We use the Adam optimizer with $learning\ rate = 0.002$ and $betas = (0.5, 0.999)$ without weight decay for training.

### D.2.1 BASELINE DETAILS

**ERM-top** $k$**.** This baseline directly models the relation between inputs and outputs using a probability distribution $p(\mathbf{y} \mid \mathbf{x})$, which is the learning target. For classification problems, when different domains

share similar frequencies during sampling, such a distribution can be approximated with the total probability formula

$$p(\mathbf{y} \mid \mathbf{x}) = \sum_{d \in D} p(\mathbf{y} \mid \mathbf{x}, d) \cdot Q(d) \approx \frac{1}{|D|} \sum_{d \in D} p(\mathbf{y} \mid \mathbf{x}, d). \tag{58}$$

For conflicting data, the resulting conditional distribution is multi-modal, and a network trained by the cross-entropy loss can provide an unbiased estimation of it. However, this method may suffer if the occurence frequencies of different domains differ significantly. Also, it introduces a hyperparameter $k$ used when selecting the top-$k$ outputs; in practice we directly use the ground-truth domain number for $k$, which leaks some additional information for this baseline.

**Semi-supervised multi-label learning.** We alternatively interpret learning from conflicting classification data as multi-label learning with missing labels: for a given input $x \in \mathcal{X}$, consider the "fully" labeled data $(x, \boldsymbol{y} = \{y_i\}_{i=1}^N)$; conflicting data can then be modeled as providing only *one* label $y \in \boldsymbol{y}$ for each $x$, with *all* other labels missing (note that this is a very extreme setting). Therefore, existing semi-supervised learning approaches may be modified to handle such problems. We consider two representative techniques, including *PseudoLabel* (Lee, 2013) and *Label propagation* (Iscen et al., 2019). Both implementations are slightly modified to fit our tasks.

*PseudoLabel* randomly allocates additional "pseudo" labels to each input to compensate the missing labels. The learning machine is trained on the augmented dataset and then reevaluate the confidence of all pseudo labels according to its predictions. All pseudo labels will iteratively be modified during training until convergence.

*Label propgation (LabelProp)* builds a graph over the examples, where each node on the graph represents a data sample, and each edge represents the distance of two nodes it collects in the feature space. The labels are propagated on adjacent nodes until all examples are fully labeled. The feature space is also iteratively adjusted during training.

**Sparse MoE.** This baseline follows the implementation in the regression task as detailed in Sec. D.1.1. Each expert is instantiated as a neural network classifier with the same architecture as the models in LEAF. We resize the one-hot labels in the examples to the same height and width as input images for the gating network.

**LEAF-EM.** We implement this variant in the same way as LEAF, expect that the allocation function is stochastic with the sampling strategy as in Sec. A.3.2.

**MLL oracle.** We provide the label annotations from all domains simultaneously for the learner, resulting in a "multi-hot" label vector for each example. This transforms the original setting to a standard multi-label learning problem. We train a single network with the same architecture as LEAF using the multi-hot label vectors as supervision by using a binary cross-entropy loss for each dimension of the multi-hot label vector.

**MTL oracle.** We provide the ground-truth domain index for each example. The raw data is then partitioned into subsets for several classification tasks without conflict, resulting in a standard multi-task learning problem. We train separate networks with the same architecture as LEAF models on these subsets. We also tried adding a shared representation layer among all networks, but found it slightly decreasing the performance, potentially due to different domains generally using distinct semantic features for their classification tasks.

# E  ADDITIONAL EXPERIMENTS AND RESULTS

In this section, we provide additional experimental results and visualization.

## E.1  EMPIRICAL ROBUSTNESS STUDY UNDER CROSS- AND IN-DOMAIN NOISE

In this section, we empirically evaluate the robustness of LEAF against both the cross- and the in-domain noise.

**Cross-domain noise.**  We consider the same setting as in our theoretical analysis in Sec. C.3, where we introduce a *mixing ratio* parameter $\alpha \in [0, 1)$ that controls the expected percentage of noisy

Table 3: Results of LEAF under different levels of cross-domain noise. We report the mean prediction errors and standard deviations on *Gender*, *Category*, and *Color* domains of *Fashion Product Images* over three independent trials.

| Mixing ratio | LEAF-EM | | | LEAF | | |
| --- | --- | --- | --- | --- | --- | --- |
| | | Error (%) | | | Error (%) | |
| | *Gender* | *Category* | *Color* | *Gender* | *Category* | *Color* |
| $\alpha = 0$ (clean) | $26.67_{\pm 3.55}$ | $6.01_{\pm 0.82}$ | $18.70_{\pm 0.92}$ | $7.87_{\pm 0.21}$ | $1.93_{\pm 0.35}$ | $12.85_{\pm 0.67}$ |
| $\alpha = 1\%$ | $27.13_{\pm 3.60}$ | $6.15_{\pm 0.94}$ | $19.20_{\pm 1.70}$ | $7.91_{\pm 0.40}$ | $1.95_{\pm 0.42}$ | $12.97_{\pm 0.88}$ |
| $\alpha = 2\%$ | $27.42_{\pm 4.10}$ | $6.23_{\pm 1.42}$ | $20.10_{\pm 3.20}$ | $7.94_{\pm 0.43}$ | $2.08_{\pm 0.49}$ | $13.22_{\pm 1.20}$ |
| $\alpha = 5\%$ | $28.70_{\pm 4.41}$ | $7.54_{\pm 1.35}$ | $25.79_{\pm 3.70}$ | $8.14_{\pm 1.12}$ | $2.18_{\pm 1.47}$ | $13.44_{\pm 1.40}$ |
| $\alpha = 10\%$ | $29.30_{\pm 3.80}$ | $6.64_{\pm 1.50}$ | $24.60_{\pm 3.60}$ | $8.17_{\pm 1.78}$ | $2.37_{\pm 2.41}$ | $13.35_{\pm 2.30}$ |
| $\alpha = 20\%$ | $29.12_{\pm 6.60}$ | $8.77_{\pm 2.70}$ | $35.73_{\pm 7.40}$ | $8.30_{\pm 2.10}$ | $2.22_{\pm 2.37}$ | $13.33_{\pm 3.05}$ |

Table 4: Results of LEAF under different levels of in-domain noise. We report the mean prediction errors and standard deviations on *Gender*, *Category*, and *Color* domains of *Fashion Product Images* over three independent trials.

| Noise ratio | | Error (%) | |
| --- | --- | --- | --- |
| | *Gender* | *Category* | *Color* |
| $\beta = 0$ (clean) | $7.87_{\pm 0.21}$ | $1.93_{\pm 0.35}$ | $12.85_{\pm 0.67}$ |
| $\beta = 1\%$ | $7.74_{\pm 0.35}$ | $2.01_{\pm 0.45}$ | $12.98_{\pm 0.97}$ |
| $\beta = 2\%$ | $12.28_{\pm 2.01}$ | $10.56_{\pm 3.22}$ | $14.00_{\pm 1.04}$ |
| $\beta = 5\%$ | $22.36_{\pm 3.87}$ | $19.72_{\pm 4.46}$ | $27.70_{\pm 3.75}$ |

examples: for each data pair in the $i$-th ($i \in [n]$) batch, with probability $1 - \alpha$ it is drawn from the sampled domain $\mathbf{d}^i = \langle P^i, c^i \rangle$, and with probability $\alpha$ it is drawn from other domains in the domain set, i.e., $D \setminus \{\mathbf{d}^i\}$. As the mixing ratio increases, the data becomes more and more "noisy" with examples from other domains. We evaluate the robustness of LEAF as well as the LEAF-EM variant on the paralleled relation setting on the *Fashion Product Images* dataset with different levels of noise: $\alpha = 0$, 1%, 2%, 5%, 10%, 20%, where $\alpha = 0$ corresponds to the standard setting without cross-domain noise. As shown in Table 3, LEAF is robust against the cross-domain noise: as the mixing ratio increases from 0 to 20%, the test prediction errors of LEAF only increases marginally. Also, LEAF is more robust to cross-domain noise than the LEAF-EM variant with generally smaller standard deviations of the prediction errors.

**In-domain noise.** We consider the setting where for each domain, every example may be randomly assigned a false label, but this false label still belongs to the label set of the domain. For instance, an image labeled as "red" in the "color" domain may be mis-labeled as another color, e.g., "yellow". We introduce a noise ratio parameter $\beta \in [0, 1)$ that controls the probability of this label noise. Like the cross-domain noise setting, we evaluate the robustness of LEAF on the paralleled relation setting on the *Fashion Product Images* dataset with different levels of noise: $\beta = 0$, 1%, 2%, 5%, where $\beta = 0$ corresponds to the standard setting without in-domain noise. As shown in Table 4, LEAF generally exhibits some performance degradation under the in-domain label noise, a similar phenomenon to the standard single-domain setting with supervised learning models; note that we do not use any additional strategy to improve the robustness of our model against the label noise, and a better result can be expected if we further adopt robust learning techniques in the literature on top of LEAF.

## E.2    MULTI-DIMENSIONAL REGRESSION ON REAL-WORLD DATA

To further demonstrate the efficacy of LEAF in regression problems, we conducted an experiment on a real-world multi-dimensional regression dataset from UCI machine learning repository: Gas sensor array under dynamic gas mixtures dataset (Fonollosa et al., 2015). This dataset contains the recordings of 16 chemical sensors exposed to two different dynamic gas mixtures and the aim is to

Table 5: Results of LEAF on multi-dimensional regression task on gas sensor array under dynamic gas mixtures dataset. We report RMSE on each domain and the macro-average RMSE over both domains.

| Methods | RMSE (domain *Methane*) | RMSE (domain *CO*) | RMSE (macro-average) |
|---------|-------------------------|--------------------|----------------------|
| ERM     | 76.5                    | 89.7               | 83.1                 |
| **LEAF** | **34.0**               | **72.6**           | **53.3**             |
| Oracle  | 31.9                    | 66.8               | 49.4                 |

predict the concentrations of gases, with 417,8504 instances and 16-dimensional attributes. We treat each gas mixture as one domain, representing Ethylene & Methane (domain *Methane*) and Ethylene & CO (domain *CO*) gas mixtures, and randomly split both domains into training (90%) and test sets (10%). Since this is a large dataset, we consider more examples in each batch and set the sampling paramters of LEAF to $n = 50$. We use a model set consisting of two neural network regressors for LEAF, and use the Adam optimizer for training.

In this task, we compare LEAF with a vanilla single model regressor (trained by ERM on the union of both domains) and an oracle regressor with two models separately trained by ERM on each domain. We report root mean square error (RMSE) on each domain and the macro-average RMSE over both domains in Table 5. The results indicate that LEAF benefits from its automatic data allocation process, surpassing the ERM baseline that only trains a single global model by a large margin and performing competitively with the oracle.

### E.3    ABLATION STUDY ON THE IMPACT OF THE NUMBER OF LOW-LEVEL MODELS AND SAMPLING PARAMETERS

In this section, we empirically study the impact of the number of low-level models and sampling parameters on the performance of LEAF, and discuss some heuristics for determining the number of low-level models in practice.

We begin by evaluating the performance of LEAF with different number of low-level models ($K = 2, 3, 4$ with $K = 3$ as the default) and sampling parameters ($m = 50, n = 2$; $m = 250, n = 1$ and $m = 250, n = 2$ with $m = 250, n = 2$ as the default) on the regression task considered in our experiment (Figure 1). As shown in Figure 6, LEAF with $K \geq R(D)$ ($K = 3$ or 4) successfully distinguishes different functions and recover each of them accurately, while LEAF with $K < R(D)$ ($K = 2$) fails, which matches the result of Theorem 2. In particular, LEAF with $K = 4$ automatically leaves one network to be redundant (dashed curve in Figure 6f), demonstrating the robustness on the number of low-level models of our framework. Meanwhile, LEAF with fewer batches $m = 50$ (Figure 6b) roughly recovers all three functions but does not yield accurate predictions in fine-grained details; this corresponds to the instance-wise estimation error term (8b) in the generalization error due to that the product $mn$ being not sufficiently large. On the other hand, LEAF with batch size $n = 1$ (Figure 6c) produces smooth predictions in the details, yet partially swaps some parts of different functions; this corresponds to the allocation estimation term (8c) in the generalization error due to that $n$ being not sufficiently large for identifying the domains.

We then empirically study the effect of additional low-level models on LEAF in the classification tasks. Concretely, for each dataset in the paralleled relation and the hierarchical relation settings, we employ a model set with the model number *larger* than the number of domains by one. We empirically observe that with an additional model, LEAF can still learn accurate predictions for each domain using a subset of the model set, with results shown in Table 6. The additional model may be simply "abandoned" as in the regression task, or becomes nearly identical to one of the other models. This naturally raises the question of **how to determine the number of low-level models in LEAF when we have no prior knowledge on the number of conflicing domains in the training data**. Here we discuss some heuristics for this problem. In summary, there are two lines of methods that can be useful:

- As shown by Theorem 2, a necessary condition of the PAC-learnability of LEAF is $K \geq R(D)$; in other words, the number of low-level models must be sufficiently large to possibly

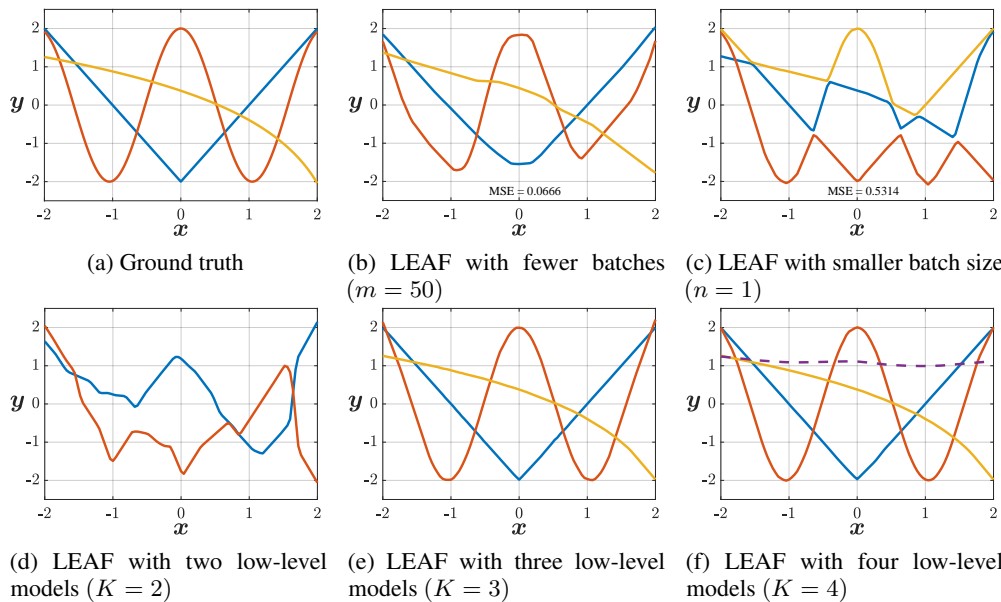

Figure 6: The impact of sampling parameters and the number of low-level models on LEAF in the regression task.

Table 6: Results of LEAF with an additional model (referred to as "LEAF-AM" in the table) on the classification tasks with paralleled and hierarchical input-output relations. We report prediction errors on *Digit* and *Color* domains of *Colored MNIST*, *Gender*, *Category* and *Color* domains of *Fashion Product Images*, and *Superclass* and *Class* domains of *CIFAR-100*.

| Methods | Colored MNIST | | Fashion Product Images | | | CIFAR-100 | |
|---------|---------------|---|------------------------|---|---|-----------|---|
| | Error (%) | | Error (%) | | | Error (%) | |
| | Digit | Color | Gender | Category | Color | Superclass | Class |
| LEAF | 1.70 | 0.03 | 7.87 | 1.93 | 12.85 | 21.40 | 25.05 |
| LEAF-AM | 2.25 | 0.17 | 7.91 | 1.82 | 13.15 | 23.52 | 24.81 |

achieve a zero training error even in the realizable case. Therefore, we can progressively increase the number of low-level models and choose the *minimal* number that elicits a low training error. This is analogous to using Akaike Information Criterion (AIC) or Bayesian Information Criterion (BIC) metrics (Claeskens et al., 2008) that both can be viewed as balancing the training error and the number of total parameters (which is directly proportional to the number of low-level models) of the whole model.

- As shown by Theorem 1, if the number of models is strictly larger than the conflict rank of the domain set, then there must exists a subset of the model set that separately captures the target functions in all domains. Therefore, we can filter out the redundant models in the model set by tracking the calling frequency of each model during training and removing the ones that are never or seldom called, or by checking the redudancy between the outputs of the models (after the models are sufficiently trained) and removing the models with identical outputs to other models for a sufficient number of examples.

## E.4 WALL-CLOCK TRAINING AND INFERENCE TIME ON THE FASHION PRODUCT IMAGES DATASET

In Table 7, we compare the computational cost of LEAF and other baselines in terms of training and inference time on the *Fashion Product Images* dataset. We measure the required wall-clock time (in seconds) for each method to reach convergence during training as well as the averaged wall-clock time for each method to predict all labels of one given input (in milliseconds). Concretely, for LEAF

Table 7: Wall-clock training and inference time of LEAF and baselines on the *Fashion Product Images* dataset. The training time of ERM-top $k$, PseoduLabel, LabelProp, Sparse MoE, LEAF-EM and LEAF is measured over 50 epochs, and the training time of MLL oracle and MTL oracle is measured over 10 epochs. The inference time of all methods is measured using an average over 3,000 test examples.

| Methods | Training time (in seconds) | Inference time (in milliseconds) |
|---|---|---|
| ERM-top $k$ | 415 | 0.67 |
| PseudoLabel | 833 | 2.00 |
| LabelProp | 915 | 0.59 |
| Sparse MoE | 886 | 0.84 |
| LEAF-EM | 672 | 0.81 |
| **LEAF** | 580 | 0.79 |
| MLL oracle | 159 | 0.68 |
| MTL oracle | 93 | 0.76 |

and all baselines except for two oracles (MLL oracle and MTL oracle), we train the models for 50 epochs with 15,000 images in every epoch; for MLL oracle and MTL oracle, we train for 10 epochs since these methods generally converge faster. For each method, we test its total inference time on the same 3,000 test examples randomly sampled from the test set and report the mean inference time on each test example. All results are obtained with PyTorch using a NVIDIA 2080ti GPU.

As shown in the table, the time cost of LEAF is generally on par with or lower than baselines that also involve iterative training (PseudoLabel and LabelProp). Although the ERM-top$k$ baseline trains faster, it learns a global model without the mechanism of data allocation and thus performs considerably worse than LEAF, as we have shown in earlier sections. Meanwhile, compared with the MTL oracle that knows the example-domain correspondences in advance, LEAF only exhibits a little additional computational overhead (note that the training time of LEAF is measured over 50 epochs, while that of MTL oracle is measured over only 10 epochs). This indicates that although LEAF incorporates an extra data allocation process implemented by the allocation function, this process only induces a limited computational cost since it only requires additional network forward processes without loss backpropagation, which is in general computational efficient.

### E.5 ADDITIONAL VISUALIZATION

In this section, we provide additional visualization on the iteration process and the low-level feature spaces learned by LEAF.

#### E.5.1 ITERATION PROCESS ON THE REGRESSION TASK

In LEAF, since the performance of the low-level models will impact the data allocation process of the high-level allocation function, the training of the allocation function and low-level networks is highly relevant. As shown by Figure 7, we provide an illustrative example in regression, in which the different colored lines refers to the allocations to different models. All networks are randomly initialized, and in each iteration, each sample may be reallocated by the allocation function and used to further train the low-level networks. With the increase of iterations, both the high-level data allocation strategy and the low-level predictions will converge with the minimization of the global training error. The last subfigure displays the final decision boundary of allocation function and indicates that all data points have been correctly allocated.

#### E.5.2 FEATURE VISUALIZATION ON THE CLASSIFICATION TASK

LEAF can extract different semantics from the same input and map them to different feature spaces by different low-level models. Figure 8 displays the features output by all low-level models learned by LEAF, where each color represents a low-level model and each point in the figure corresponds to an input image in the *Fashion Product Images* dataset. As shown in the figure, different low-level models learn metrics based on different semantics of the inputs.

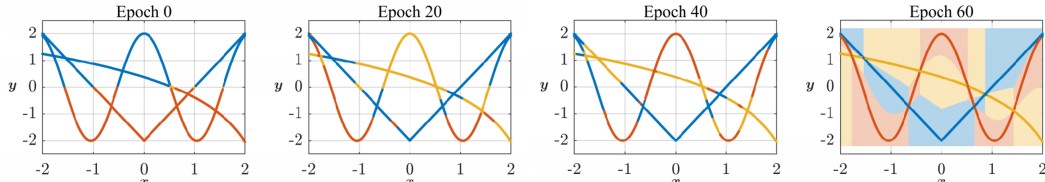

Figure 7: An example of the iteration process and the final decision boundaries of the allocation function in the regression task.

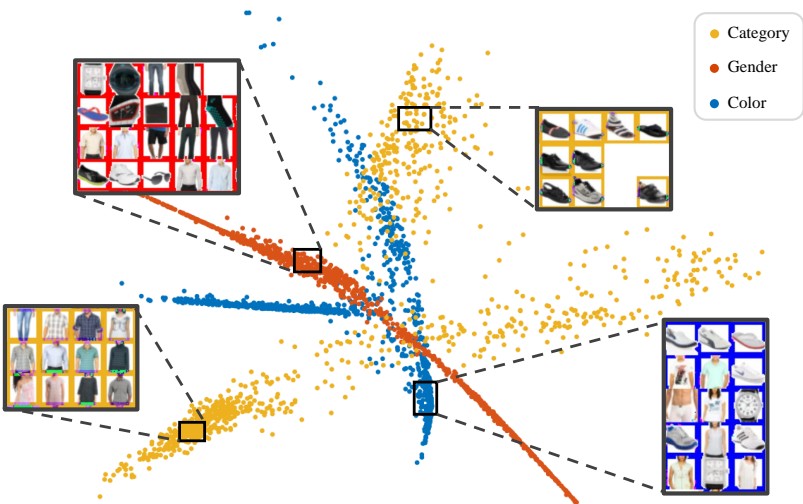

Figure 8: Additional visualization of different low-level feature spaces of LEAF on the *Fashion Product Images* dataset.

## F    MORE DISCUSSION ON RELATED WORK

In this section, we provide more discussion on related work.

**Ensemble learning.**    Ensemble learning approaches typically employ multiple models to cooperatively solve a given task (Dietterich, 2002; Zhang & Ma, 2012; Sagi & Rokach, 2018; Zhou, 2021). The prediction of each model is combined by weighting (boosting), majority voting (bagging) or learning a second-level meta-learner (stacking). Since different models process the same set of data (although sometimes with different instance weights), there is typically no explicit "hard" allocation process in ensemble learning between the examples and the models. In contrast, the multi-model architecture of LEAF is driven by the inherent conflict in the conflicting training data, and each model only handles a proportion of the whole dataset without overlapping.

**Domain adaptation and domain generalization.**    Domain adaptation (Ben-David et al., 2010a; Pan & Yang, 2010; Tan et al., 2018; Wang & Deng, 2018; Hoffman et al., 2018) and domain generalization (Blanchard et al., 2011; Muandet et al., 2013; Zhou et al., 2021) consider the scenario where the learner trained on one or multiple source domain(s) is transferred to one or multiple new target domain(s). Typically, domain adaptation focuses on the problem where there exist some labeled or unlabeled instances in the new domain, while domain generalization considers the setting where there the information of the new domains is inaccessible during training (i.e., zero-shot generalization). In other words, these formulations focus on the *adaptation* or *generalization* capability of the model on the target domain(s) and do not consider training on the sources domains *itself* to be an issue. By contrast, LEAF focuses on the multi-domain *training* process and considers the scenario where directly training a global model by ERM using the data from multiple conflicting domains is problematic, and aims to resolve this training issue by performing automatic data allocation. In other words, the data setup between our work and unsupervised domain adaptation is fundamentally different: a common

assumption adopted by most of unsupervised domain adaptation is covariate shift assumption (Ben-David et al., 2010b), i.e., the input-conditioned label distribution $p(\mathbf{y} \mid \mathbf{x})$ is the same for all domains while the input marginal distribution $p(\mathbf{x})$ varies. By contrast, we define conflicting domains as the target function from different domains give distinct outputs for the same set of inputs, which stands for varying $p(\mathbf{y} \mid \mathbf{x})$. Therefore, the setup of unsupervised domain adaptation and our work are orthogonal. We also note that from a theoretical view, existing domain adaptation methods are provably infeasible in our problem since their success requires the existence of a model with low error simultanesouly on all domains (see e.g., Mansour et al. (2009); Ben-David et al. (2010b)), which is not true if varies a lot across different domains. Finally, the example-domain correspondences are usually available in domain adaptation and domain generalization settings, while LEAF weakens this assumption by only assuming that the examples in each batch are obtained from the same domain.

**Latent variable modeling.** Latent variable modeling (Skrondal & Rabe-Hesketh, 2004; Muthén, 1989; 2002) is a general approach that models the relation between observed variables with unobserved, latent variables. In a broad sense, LEAF falls into the category of latent variable modeling by treating the domain of each data pair as a (discrete) latent variable. Nevertheless, the notion of latent variable modeling itself is rather general, and distinct methods may be developed based on the specific problem setup. To the best of our knowledge, the problem of learning from conflicting data has not been formalized or addressed by prior latent variable modeling methods. Meanwhile, several key design choice (using Viterbi EM rather than standard EM) and data assumption ($n > 1$) of LEAF is tailored to our problem setup, thus differing LEAF from general latent variable modeling approaches. See the next section for a more detailed discussion.

## G NECESSITY OF USING BATCHES OF DATA ($n > 1$) RATHER THAN SINGLE DATA PAIRS ($n = 1$) FOR CONFLICTING DOMAIN IDENTIFICATION

In this section, we discuss the *necessity* of the assumption that we observe a size-$n$ batch rather than single data pairs ($n = 1$), which is often considered by general latent variable modeling methods without further structural assumptions on data. To summarize, $n = 1$ **poses fundamental difficulty in identifying conflicting domains and, in the worst case,** $n = 1$ **makes identification of conflicting domains impossible.** We will first construct two examples, one for regression and one for classification, to illustrate this impossibility:

- Consider a regression problem with two domains, namely $d_1$ and $d_2$, where training $(x, y)$ pairs are uniformly sampled from $y_1 = x, x \in [0, 1]$ in $d_1$ and $y_2 = 1 - x, x \in [0, 1]$ in $d_2$, respectively. Then, in the $n = 1$ setting, both of the following two solutions achieve zero training error:

  **S1:** $h_1(x) = x, \ x \in [0, 1]$ and $h_2(x) = 1 - x, \ x \in [0, 1]$;

  **S2:** $h_1(x) = \begin{cases} x, \ x \in [0, 0.5) \\ 1 - x, \ x \in [0.5, 1] \end{cases}$ and $h_2(x) = \begin{cases} 1 - x, \ x \in [0, 0.5) \\ x, \ x \in [0.5, 1] \end{cases}$.

  However, only **S1** is the desired solution that separately recovers $y_1$ by $h_1(x)$ and recovers $y_2$ by $h_2(x)$, while **S2** does not recover either $y_1$ or $y_2$ by any of the model. However, **S1** and **S2** are not identifiable when $n = 1$ since both of them are optimal during training. We can also see that when $n > 1$ would effectively present the undesired solution **S2**, since we may sample two points $(x^1, y^1)$ and $(x^2, y^2)$ from the same domain with $x^1 \in [0, 0.5)$ and $x^2 \in [0.5, 1]$ in the same batch, so that **S2** does not achieves zero training error anymore. In fact, we did observe such cases in our regression experiments (see Figure 6c in Appendix E.3), where $n = 1$ leads to incorrect identification of domains.

- Consider a classification problem that follows the CIFAR-100 setup in our experiments, where each $x$ corresponds to the input image and $y$ corresponds to the label. There are two domains, $d_1$ and $d_2$, where images in $d_1$ are labeled by superclass names, while images in $d_2$ are labeled by class names. As an illustrative example, we consider $d_1$ with labels "fish" and "flowers" and $d_2$ with labels "shark", "flatfish", "roses" and "sunflowers", with the first two labels belonging to "fish" and the last two labels belonging to "shark". Both $d_1$ and $d_2$ contains input images that belong to "shark", "flatfish", "roses" and "sunflowers" with the same distribution. Then, in the $n = 1$ setting, both of the following two solutions achieve

Table 8: Comparisons between LEAF, LEAF-EM and EM ($n = 1$) on CIFAR-100. We report prediction errors *Superclass* and *Class* domains.

| Methods | *CIFAR-100* | |
|---|---|---|
| | Error (%) | |
| | *Superclass* | *Class* |
| LEAF | 21.40 | 25.05 |
| LEAF-EM | 32.71 | 33.83 |
| EM ($n = 1$), 1st run | 61.71 | 52.46 |
| EM ($n = 1$), 2nd run | 67.43 | 55.20 |
| EM ($n = 1$), 3rd run | 71.33 | 48.79 |

zero training error. **S1:** model $h_1$ classifies all images by their superclass labels, while model $h_2$ classifies all images by their class labels. **S2:** model $h_1$ classifies fish images according to their superclass labels and classifies flower images by their class labels, while model $h_2$ classifies fish images by their class labels and classifies flower images by their superclass labels. In other words, **S1** recovers the target function in both domains, while **S2** fails to recover any target function since the outputs of both models $h_1, h_2$ in **S2** are *mixtures* of superclass labels and class labels. Similar to the regression case, $n > 1$ can prevent **S2** from being an optimal training solution since superclass labels "fish" and "flowers" can be sampled in the same batch and thus should be predicted by the same model.

Besides the concrete failure cases we provide above, *our theoretical result on the generalization error bound of LEAF also sheds light on the failure of the $n = 1$ case:* to bound the generalization error between empirical and expected allocation functions (Eq. (8c) in Theorem 3), we generally require a sufficiently large $n$. Therefore, $n = 1$ may create a non-negligible gap between empirical (training) error and expected error. Since our Theorem 1 suggests an equivalence between domain identification and reaching a small *expected error*, having a gap between empirical and expected errors can pose difficulty in domain identification.

Finally, we run extra experiments to illustrate this failure of $n = 1$ on CIFAR-100: we added a new baseline denoted by *EM ($n = 1$)* that (1) uses standard EM and (2) eliminates the information about which datapoints come from the same batch. We evaluated its performance on the CIFAR-100 dataset (with two low-level models), following the same setup as in our main experiments in the main text. However, this baseline cannot converge to stable solution, and we found its performance have large fluctuations across different runs and generally much worse than both *LEAF* and *LEAF-EM*. This validates the counterexample on the identifiability issue in the $n = 1$ case in the above. Detailed results are in Table 8.

