# OpenReview forum: "Learning from conflicting data with hidden contexts"
_ICLR.cc/2023/Conference — Submitted to ICLR 2023_

### Official Review · Reviewer_4gxX · 2022-10-23

**Confidence:** 3
**Correctness:** 3
**Technical Novelty And Significance:** 3
**Empirical Novelty And Significance:** 3
**Recommendation:** 6

**Clarity, Quality, Novelty And Reproducibility:**

I think the writing should be improved. Even the problem setup is not totally clear to me. According to the PAC learnability analysis in the appendix, the authors consider something similar to the top-k error, but there is no discussion about related work on the top-k error.

**Strength And Weaknesses:**

# Strengths
- The paper seems to tackle an important and interesting problem.
- The method performs excellently according to the experiments.

# Weaknesses
- The problem setup is not very clear. In particular, the evaluation metric, or the test error, is not clearly defined, and the paper does not explain what information is given to the model in prediction.
- It does not seem easy to choose $K$. The authors mention some connection to $R(D)$, but how do we estimate this number?
- In Eq. (3), the method allows $\hat{g}$ to look at $y^{ij}$, which feels strange to me because we don't have access to it in the test phase.
- There is no guarantee that the proposed training minimizes the
- Knowing each set of $m$ samples comes from one distribution seems to be a strong assumption. The theory requires $m$ tends to infinity to ensure the convergence of $\hat{\operatorname{err}}(H)$ to $\operatorname{err}(H)$. The paper briefly mentions this point, but the argument is not very convincing to me.

**Summary Of The Paper:**

This paper proposes a method for learning from data following a mixture of distributions whose input-output relationships are different. The proposed method maintains multiple hypothesis functions accounting for those different distributions and minimizes the loss between the label and the closest output among those given by the hypothesis functions. The authors explain a connection between the proposed method and the EM algorithm. They also show an identifiability result and a generalization error bound between the proposed training objective and an oracle risk. The experiments show the superiority of the proposed method against other methods that are not designed for this setup.

**Summary Of The Review:**

I like the direction of the work, but I think the quality of the paper should be improved before being published. There are many major issues as I mentioned in the weakness section. I suggest rejecting the paper.

---

> ### Author Response · Authors · 2022-11-11
> **Response to Reviewer 4gxX (3/3)**
>
>
> >Knowing each set of $m$ samples comes from one distribution seems to be a strong assumption. The theory requires $m$ tends to infinity to ensure the convergence of $\hat{er}(H)$ to $er(H)$. The paper briefly mentions this point, but the argument is not very convincing to me.
>
> - While our generalization error bound in Theorem 3 requires $n\to\infty$ (we used $n$ to denote the number of examples in one batch, and used $m$ to denote the number of batches in our paper) for $\widehat{er}(H)\to er(H)$, we note that **this bound should be interpreted as characterizing the _worst case_ generalization error**. In other words, since _we do not make any additional assumptions on domains_ in Theorem 3, its upper bound corresponds to the extreme case which may not exactly align with our experiments. For example, one can construct a scenario where there are two domains whose target functions are the same on most of the support while only differing on a very small subset of inputs. This would make identifying them inherently difficult and thus requires a large $n$ to ensure that the inputs on which two domains conflict are sampled with a sufficiently large probability. However, when different domains conflict on most of $x$, identifying conflicts would be much easier and thus requires a smaller $n$. _Empirically, in our experiments we found that $n<5$ suffices on all tasks. We believe that by introducing reasonable assumptions on the relation between domains (e.g., the ratio of conflicting examples over all examples for each domain pair), we can obtain a tighter generalization error bound with respect to $n$_. However, this refinement is rather incremental compared to our main contribution and thus we leave this extension to future work.
>
> - In addition, we have empirically shown that **our method is robust to up to 20% cross-domain noise in data batches (see Table 3 in Appendix E.1)**, which further makes our assumption of $n>1$ more realistic since in practice we do not require all $n$ examples in a batch to be drawn from exactly the same domain for our method to work.
>
> - Finally, we can construct counterexamples to show that $n=1$ can lead to identifiability issues on conflicting domains, thus it is actually a **necessary** assumption in our problem setup (see our response to Reviewer zrsD as well as Appendix G in the revised paper for more details).
>
> >According to the PAC learnability analysis in the appendix, the authors consider something similar to the top-k error, but there is no discussion about related work on the top-k error.
>
> Our work is **essentially different** from methods that optimize the top-$k$ error of the classifier. Notably, **the training objective of LEAF involves learning $K$ models simultaneously, instead of learning a _single_ model by optimizing its top-$K$ error**. Our PAC learnability analysis extends conventional PAC analysis by considering learning $K$ target functions simultaneously rather than learning only _one_ target function (with or without top-$k$ error), thus it also does **not** "consider something similar to the top-$k$ error".
>
> ---
>
> **Please let us know if our explanations above are still unclear, or if you have any other questions or concerns. Thank you again for your review!**

---

> ### Author Response · Authors · 2022-11-11
> **Response to Reviewer 4gxX (2/3)**
>
> >It does not seem easy to choose $K$. The authors mention some connection to $R(D)$, but how do we estimate this number?
>
> **We do have discussed some practical methods for estimating the number of low-level models $K$ in Appendix E.3, and explicitly referred to this discussion in Section 6 in the main text**. Here we summarize the discussion on this topic in Appendix E.3. In general, there are two lines of methods that can be useful:
>
> - As shown by Theorem 2, a necessary condition of the PAC-learnability of LEAF is $K\ge R(D)$; in other words, the number of low-level models must be sufficiently large to possibly achieve a zero training error even in the realizable case. Therefore, we can progressively increase the number of low-level models and choose the _minimal_ number that elicits a low training error. This is similar to using metrics such as Bayesian Information Criterion (BIC) that balances the training error and the number of total parameters of the model.
>
> - As shown by Theorems 1 and 2, if the number of low-level models $K$ is not smaller than $R(D)$, then there must exist a _subset_ of the model set that separately captures the target functions in all domains, and we can thus proceed by _filtering out_ the "redundant models" in the model set. Compared with the first method, this method would be less costly since it does not require retraining. It also does not require a heavy hyperparameter tuning on the number of low-level models as in general one only needs to ensure that the number of models is large enough, rather than being strictly equal to the ground-truth of $R(D)$. In practice, different heuristics can be employed in this filtering process, such as:
>
>   + We can track the calling frequency of each model during training and remove the ones that are never or seldom called.
>   + We can also examine the similarity between the outputs of the models (after the models are sufficiently trained) and remove the models with nearly identical outputs to other models for a sufficient number of examples.
>
> In Appendix E.3, we report empirical results on classification tasks where the number of low-level models is _larger_ by one than the ground-truth of $R(D)$ (Table 6). The results show that there indeed exists a subset of models that yields comparable performance in all tasks, compared with the ideal case where the number of low-level models is set accurately. This corroborates Theorems 1 and 2 and indicates that our method is robust to the under-specification of the number of low-level models.
>
>
> >In Eq. (3), the method allows $\hat{g}$
> to look at $y^{ij}$, which feels strange to me because we don't have access to it in the test phase.
>
> - Indeed, **LEAF does not have access to the ground-truth labels in the test phase**. As we have mentioned in our response to your former concerns, our evaluation metrics only require the **low-level model set** of LEAF, which does **not** take $y$ as input.
>
> - However, during training, it is necessary to let our **allocation function** $\hat{g}$ (not the low-level models) have access to $y^{ij}$ since **it is not possible to perform data allocation with only $x^{ij}$ on conflicting data**: as in Definition 1, the crux of the conflict between domains is that different domains label a **same** set of $x$ **differently**, thus $x^{ij}$ itself without $y^{ij}$ is insufficient to decide whether domains are conflicting.
>
> >There is no guarantee that the proposed training minimizes the
>
> **Unfortunately the sentence in the original review is incomplete, and thus we are not able to give a very accurate response.** We hypothesize that you were probably saying that there is no convergence analysis on our EM-style training process and discuss this point in the following.
>
> Since our method is derived from an EM framework, the existing convergence guarantee of (generalized) EM methods also applies to our setting. Notably, it is known that under fairly general conditions on the monotonicity of the likelihood with respect to posterior $q$ in each iteration, the likelihood values in (generalized) EM converge to stationary points with zero gradients (see e.g., Section 3.1 of [1]). Therefore, our training scheme also shares a similar guarantee, i.e., converging to a stationary point of the empirical error of LEAF. We have added this discussion to Section 2.3 and included more technical details in Appendix A.4 in the revised paper.
>
> [1] McLachlan et al. The EM algorithm and extensions. 2007.

---

> > ### Comment · Reviewer_4gxX · 2022-11-18
> > **Additional comments**
> >
> > About the heuristics for choosing K, have the authors tried them in experiments? Since they are heuristics, there is no way to evaluate them without seeing experimental results.
> >
> >
> > > Indeed, LEAF does not have access to the ground-truth labels in the test phase. As we have mentioned in our response to your former concerns, our evaluation metrics only require the low-level model set of LEAF, which does not take as input.
> >
> > I still do not see how one can make a prediction without the label in the test phase.
> >
> >
> > >>There is no guarantee that the proposed training minimizes the
> > >
> > > Unfortunately the sentence in the original review is incomplete, and thus we are not able to give a very accurate response. We hypothesize that you were probably saying that there is no convergence analysis on our EM-style training process and discuss this point in the following.
> >
> > I apologize for my incomplete sentence. Yes, it was about the EM algorithm but not about its convergence. My concern was about the quality of the solution after the convergence because it can only find a local minimum if I'm not mistaken.

---

> > > ### Author Response · Authors · 2022-11-19
> > > **Response to additional comments**
> > >
> > >
> > > Thank you for your additional comments! We address your follow-up concerns below:
> > >
> > > >About the heuristics for choosing K, have the authors tried them in experiments? Since they are heuristics, there is no way to evaluate them without seeing experimental results.
> > >
> > > Yes! We have tested them in the experiments. For example, the following table shows the **training error** (measured by the cross-entropy loss) on the Colored MNIST (two conflicting domains) and Fashion Product Images (three conflicting domains) datasets with different number of low-level models $K$:
> > >
> > > |$K$|Training error on Colored MNIST|Training error on Fashion Product Images|
> > > |:--:|:--:|:--:|
> > > |1|0.69 $\pm$ 0.03| > 1.80|
> > > |2|0.16 $\pm$ 0.01| 1.64 $\pm$ 0.01|
> > > |3|0.15 $\pm$ 0.01| 1.38 $\pm$ 0.05|
> > > |4|--|1.35 $\pm$ 0.03|
> > >
> > > As shown in the table, _by progressively increasing the number of $K$, the training error first exhibits a rapid decrease and then plateaus_ (after reaching $K=2$ on Colored MNIST and $K=3$ on Fashion Product Images). We can accordingly select an appropriate $K$ for different datasets (2 for Colored MNIST and 3 for Fashion Product Images). Note that the training set of Fashion Product Images is itself a bit of noisy, so we cannot obtain a very small training error even with sufficient number of low-level models as in Colored MNIST; however, the general trend of training error is the same.
> > >
> > > We will include this result and other related empirical results on validating the methods of choosing $K$ in the revised version of our paper.
> > >
> > > >I still do not see how one can make a prediction without the label in the test phase.
> > >
> > > Hope that our response in the former comment already addressed your concern here. To recap, the reported results in the current version of our paper is based on the multi-label prediction for each unlabeled test example by calling each model in the model set and aggregating their outputs; in addition, we can use a small subset of labeled test examples to select a model from the model set and then use it to predict other unlabeled test examples when we know that the test examples come from the same domain.
> > >
> > > >My concern was about the quality of the solution after the convergence because it can only find a local minimum if I'm not mistaken.
> > >
> > > Thank you for your question! In fact, the previous version of our paper (before the rebuttal revision) included a paragraph discussing the convergency of our training procedure. In the last revision, we removed this paragraph due to the space limit. However, your concern made us feel that this discussion is indeed important and therefore we plan to add it back in.
> > >
> > > You are right that EM does not theoretically guarantee the convergence toward the global minimum. Therefore, it is natural to wonder whether our algorithm is vulnerable to the local optima of data allocations during optimization. However, we did not observe this in practice. We believe that _an important reason for this lies in the property of conflicting data itself_: consider a classification problem with conflicting data, a model trained on some examples provably gives **worse-than-chance** (i.e., even worse than a random-initialized model) performance in expectation on the examples conflicting with those which the model is trained on. Therefore, conflicting examples are not likely to be assigned to the same model consistently during the training process as long as the models have sufficient capacity (e.g., when we use deep neural networks). Nevertheless, it is very hard to rigorously prove this result since it requires analyzing the non-linear and non-convex optimization process of neural networks, and thus is beyond the scope of this paper.

---

> ### Author Response · Authors · 2022-11-11
> **Response to Reviewer 4gxX (1/3)**
>
> Thank you for your feedback! We are glad to hear that you found our problem setup important and interesting. In this rebuttal, we will do our best to address your concerns. As requested, we have also revised the paper to improve its clarity, and the details of the revision can be found in our response to your detailed concerns below. _We hope that this response can increase the score in your rating._
>
> In the following, we provide point-by-point responses to your concerns:
>
> >The problem setup is not very clear. In particular, the evaluation metric, or the test error, is not clearly defined, and the paper does not explain what information is given to the model in prediction.
>
> While we do not give a mathematical definition of the test error in the problem formulation section, **we do explicitly introduce our evaluation metric in our experiments (see "Evaluation protocol" paragraphs in Section 4)**.
>
> - For the regression task, we plot the fitted curves of all low-level models of LEAF and compare them visually with groud-truth functions; since this task is mostly illustrative, we do not apply formal metrics such as MSE, although such a metric can be similarly defined as in classification.
>
> - For classification tasks, we report the errors based on the low-level model that yields the _smallest_ error for each domain. Formally, for domain $d$ with test examples $(x_1,y_1),\ldots,(x_{n_d},y_{n_d})$, the error of LEAF is defined as $\min_{h\in H} \frac{1}{n_d} \sum_{i=1}^{n_d} \mathbf{1}[h(x_i)\ne y_i]$ with $\mathbf{1}$ being the indicator function. It is easy to see that this is an unbiased estimator of the expected error $er(H)$ in Definition 3 as $n_d\to\infty$.
>
> It is worth noting that in all of our experiments, **all low-level models of LEAF only have access to the inputs and do _not_ have access to the ground-truth labels at test time**.
>
> In the revision, we have rephrased and expanded the "Evaluation protocol" paragraph in our classification experiments (Section 4.2) to improve its clarity, incorporate the formally defined evaluation metric and underscore the fact that we do not use ground-truth labels as inputs at test time.

---

> > ### Comment · Reviewer_4gxX · 2022-11-18
> > **Evaluation metric**
> >
> > Thanks for the response.
> >
> > I know the experiment part had some explanation about the evaluation, but it is not satisfactory. The evaluation metric is a core part of the goal of a learning problem because otherwise we cannot even define generalization. I recommend clearly defining the goal of the problem including the evaluation metric at the level of the problem setup, before talking about methods or experiments.
> >
> > Also, the metric described in the experiment part, $\min_{h\in H} \frac{1}{n_d} \sum_{i=1}^{n_d} 1[h(x_i) \neq y_i]$, does not reflect a practical situation because we would need access to the label $y_i$ in order to determine the best hypothesis $h$. How would we make a prediction without the label $y_i$ in practice?

---

> > > ### Author Response · Authors · 2022-11-19
> > > **Thank you for your feedback! Author response**
> > >
> > >
> > > Thank you very much for your engagement in discussion and continual feedback! We very much value your recommendation of defining the evaluation metric in the problem setup and will revise the corresponding section in a future version of the paper (unfortunately we are not able to upload a revised paper now due to the timeline of ICLR).
> > >
> > > - In the current version of our paper, we (informally) describe the goal of our setup at the beginning of problem formulation (Section 2.1): "**the goal of the learner is to learn all target functions $\\{c_i\\}_{i=1}^N$**". Then, after formally introducing the sampling process and the training objective, we show by Theorem 1 that **this goal is equivalent to minimizing the expected error $er(H)$ (Eq. (4))** (with our choice of the allocation function). This is why our generalization error analysis is based on bounding the difference between $er(H)$ and $\widehat{er}(H)$: the reason is that _minimizing the expected error $er(H)$ indeed captures our ultimate goal of learning all target functions from potentially conflicing domains simultaneously_, thus bounding the generalization error between empirical and expected errors would be meaningful towards our goal. Nevertheless, your comment has reminded us that this connection is kind of elusive in the current paper, and we will rephrase the corresponding paragraphs to make this more clear in the revision.
> > >
> > > - Another reason for not explicitly opting for a specific evaluation metric in the problem setup is, **given a learned model set, we may evaluate it in different ways that reflect different practical settings, which correspond to different forms of evaluation metrics (see details below).** Note that this is common in many machine learning problems, e.g., we can evaluate the performance of a classifier by its accuracy, precision, recall or F1 score on the test set, depending on the problem of interest. On the other hand, all of these evaluation metrics are closely connected to the theoretically defined value of _population risk (expected risk)_, and bounding the difference between the population risk and the empirical risk (i.e., the generalization error) constitutes a major part of statistical learning theory. In other words, generalization error analysis does not require the exact equivalence between the expected error and the evaluation metric; in practice it only requires the expected error to be a suitable proxy of the error based on the evaluation metric to ensure that the analysis is meaningful.
> > >
> > > **On the evaluation process (metrics) of LEAF:** The evaluation metric $\min_{h\in H} \frac{1}{n_d} \sum_{i=1}^{n_d} \mathbf{1}[h(x_i)\ne y_i]$ reflects the practical setting where **we have no prior knowledge of the test domain and thus invoke _every_ model in the model set for each unlabeled test example**. As a concrete example, consider the Colored MNIST classification task in our experiment where each digit has a color label and a digit label, and LEAF learns a model set with two models, one for color prediction and one for digit prediction. Then, when an unlabeled test example arrives, it is **not possible** to judge whether "color" or "digit" gives the "right" label without further prior knowledge of the task of interest. Therefore, _we call **both** models for this test example to give it a color label and a digit label simultaneously, which is arguably a reasonable choice since both labels equally make sense_. **The above evaluation metric represents the performance of this natural "multi-label" evaluation process:** to predict **all possible labels** for each example correctly, one must learn a model in the model set $H$ for **each** target function, resulting in a small error on this metric; by contrast, if a target function is not properly learned, then **all** models in $H$ would perform badly on this domain, yielding a large evaluation error.
> > >
> > > - Depending on the practical setting, we may also evaluate the performance of LEAF with different metrics. For example, in practice if we know that all test examples come from the **same** domain and also have a small set of **labeled** examples from this domain, we can use this set of labeled examples to choose a model (the one that yields the smallest error on this example set) from the model set and then use the chosen model to predict **other unlabeled test examples**. We did not report the performance under this evaluation metric separately since in our tasks typically $<0.1\\%$ labeled examples from the test domain suffice for choosing the correct model, which makes the reported domain-wise error of LEAF essentially similar to the ones that we have reported in the current paper.
> > >
> > > We hope that this discussion is clear and addresses your remaining concerns on the evaluation process of our method. We look forward to your response.

---

> > > > ### Comment · Reviewer_4gxX · 2022-11-20
> > > > **Practical implication of the oracle**
> > > >
> > > > Thank you for the response.
> > > >
> > > > > As a concrete example, consider the Colored MNIST classification task in our experiment where each digit has a color label and a digit label, and LEAF learns a model set with two models, one for color prediction and one for digit prediction. Then, when an unlabeled test example arrives, it is not possible to judge whether "color" or "digit" gives the "right" label without further prior knowledge of the task of interest. Therefore, we call both models for this test example to give it a color label and a digit label simultaneously, which is arguably a reasonable choice since both labels equally make sense. The above evaluation metric represents the performance of this natural "multi-label" evaluation process: to predict all possible labels for each example correctly, one must learn a model in the model set for each target function, resulting in a small error on this metric; by contrast, if a target function is not properly learned, then all models in would perform badly on this domain, yielding a large evaluation error.
> > > >
> > > > I feel the example might be a little misleading because the models do not provide the semantic information about whether the predictions are about "color" or "digit". They are encoded in the same space of context-free labels, and we don't know which model to choose even if the context is revealed in the test phase.
> > > >
> > > > What I am not comfortable with about the paper is that it does not discuss the practical implication of the fact that the evaluation metric implicitly assumes the existence of the oracle that chooses the right model for us from the multiple hypotheses learned by the method (i.e., the "expected allocation function" or the min operation over the hypotheses of the metric for the experiment). I would like to see a practical situation in which we really have that oracle, or having multiple predictions is already useful.
> > > >
> > > > > Depending on the practical setting, we may also evaluate the performance of LEAF with different metrics. For example, in practice if we know that all test examples come from the same domain and also have a small set of labeled examples from this domain, we can use this set of labeled examples to choose a model (the one that yields the smallest error on this example set) from the model set and then use the chosen model to predict other unlabeled test examples.
> > > >
> > > > I indeed find that kind of evaluation better in some sense because a method can fool the sample-wise metric $\min_{h\in H} \frac{1}{n} \sum_{i=1}^{n} 1[h(x_i) \neq y_i]$ by using $|\mathcal{Y}|$ constant hypotheses that output different values of $\mathcal{Y}$, without performing any learning.

---

> > > > > ### Comment · Reviewer_4gxX · 2022-11-20
> > > > > **One practical example mentioned by the authors**
> > > > >
> > > > > This response from the authors on another thread makes sense to me:
> > > > > > we can use a small subset of labeled test examples to select a model from the model set and then use it to predict other unlabeled test examples when we know that the test examples come from the same domain
> > > > >
> > > > > Hopefully, the authors have more practical examples. I think it would be really nice to have such a discussion in the paper. I can raise my score if the authors agree on this point.

---

> > > > > > ### Author Response · Authors · 2022-11-21
> > > > > > **We very much agree with your suggestion; more practical examples**
> > > > > >
> > > > > > Thank you for your continual engagement! **We are very glad to hear that we have reached a consensus on the evaluation and would be very happy to add the discussion on the practical implications of the evaluation metrics in the revised version of our paper.** Below we give more practical examples on these points:
> > > > > >
> > > > > > - **"Select-and-predict evaluation":** In many practical scenarios, we do have access to a small subset of labeled examples at test time for us to perform model selection from the learned model set. For example,
> > > > > > consider training a language model that predicts the next word in a mobile keyboard. Since people from different populations (a context that is often hidden since many public datasets do not have relevant auxiliary attributes) are likely to use language differently, training on the data collected from multiple populations using LEAF is likely to yield multiple models that suit different populations. At test time, by monitoring the accuracy of these models on a particular device, one can decide to which model this mobile user belongs, which can be viewed as a form of on-device personalization. A similar example is also discussed in [1] (Section 3.3.1) in the context of federated learning.
> > > > > >
> > > > > > - **"Multi-label evaluation":** There are also practical situations in which making "multi-label" predictions by calling all models for each example is useful. This evaluation protocol is similar to the protocol of **multi-label learning** [2], where each input is assigned a _label set_ consisting of _multiple_ labels in the label space. Thus, the learner that predicts the correct label set for each test input achieves the highest performance with respect to the evaluation metric. For example, given a video clip filmed in New York, a model that tags it by the label set of {"building", "urban", "New York"} would be more informative to the user than a model that only predicts a single label "building".
> > > > > >
> > > > > >   - It is worth noting that the trivial solution consisting of $|\mathcal{Y}|$ sub-models that constantly output different labels in the label space **cannot** fool our metric $\min_{h\in H} \frac{1}{n_d} \sum_{i=1}^{n_d} \mathbf{1}[h(x_i)\ne y_i]$. This is because **the $\min$ operator in the metric is defined over the _empirical mean error_ of each domain rather than the error of each test example**. Hence, this trivial solution does not give a small evaluation error unless all test examples in each domain are with the same label.
> > > > > >
> > > > > > [1] Kairouz et al. Advances and open problems in federated learning. Foundations and Trends® in Machine Learning, 2021.
> > > > > >
> > > > > > [2] Zhang et al. A review on multi-label learning algorithms. IEEE transactions on knowledge and data engineering, 2014.

---

> ### Author Response · Authors · 2022-11-17
> **Looking forward to your feedback!**
>
> Dear Reviewer 4gxX,
>
> Thank you again for your detailed review and constructive comments. Since the official discussion period is ending soon, we would like to kindly remind you to check our response and the revised paper. Any feedback would be appreciated. Thank you!

---

### Official Review · Reviewer_eW7y · 2022-10-24

**Confidence:** 4
**Clarity, Quality, Novelty And Reproducibility:** The method is clearly described in th…
**Correctness:** 3
**Technical Novelty And Significance:** 3
**Empirical Novelty And Significance:** 3
**Recommendation:** 8

**Strength And Weaknesses:**

- I think the problem of conflicing contexts seems novel and important for solving the real-world problems.
- The examples for context-conflicting cases are well exampled.
- The LEAF method looks sound and novel enough.
- Experimental results are impressive, showing superior performance to several competitive baselines (ie. meta-learning, MoE, etc)
- Theoretical bounds also look sound.

**Summary Of The Paper:**

Authros proposed the method that can deal with conflicting contexts scenario. Authors proposed a LEAF architecture for solving the problem. Authors also showed the effectiveness of the proposed method using toy and real-world datasets. Also, they offer the theoretical bounds for their method.

**Summary Of The Review:**

I think the draft contains the solid method for solving unique aspect of the supervised learning task. The problem looks important and the proposed method looks also sound. Experimental and theoretical analyses are also thorough. I recommend accept for this draft.

---

> ### Author Response · Authors · 2022-11-11
> **Response to Reviewer eW7y**
>
> Thank you for your positive review! We appreciate your feedback and would be happy to engage in any further discussion on our work during the rebuttal phase.

---

### Official Review · Reviewer_yevj · 2022-10-25

**Confidence:** 3
**Correctness:** 4
**Technical Novelty And Significance:** 3
**Empirical Novelty And Significance:** 3
**Recommendation:** 8

**Clarity, Quality, Novelty And Reproducibility:**

- The paper is of high quality, overall rigorously written and easy to follow.
- The problem setting has reasonably high novelty and pratical importance.
- Reproducibility is guaranteed by the detailed description and provided sample python code.


**Strength And Weaknesses:**

Strength
- This paper considers the new problem setting of learning from hiddenly conflicting data, and proposes a reasonable rigorous definition based on domain generalization.
- It then proposes a pratical EM-based algorithm for the new problem, and analyzed its learnability and generalization ability compared with existing general supervised learning case.
- Empirical evaluation is carefully conducted and experiment python files are provided with detailed README.

Weakness
- As the proposed algorithm is based EM, it is nice to have dicussions on its convergence conditions.
- Hidden conflicts defined in this paper resembles graphical models with a latent variable, although graphical models cast stronger assumptions on the data generation process. Some dicussion on this topic is desired.


**Summary Of The Paper:**

This paper first formally defines a new and practically important problem setting where data are generated from multiple domains and the generation process is completely hidden. It then propose a concrete EM-based algorithm with reasonable theoretical guarantees. Empirical evaluations are conducted on various tasks and high quality python code is provided.


**Summary Of The Review:**

I recommend to accept the paper considerting the overall contribution of problem setting, algorithm, theoretical analysis and empirical evaluation.

---

> ### Author Response · Authors · 2022-11-11
> **Response to Reviewer yevj**
>
> Thank you for your positive review! We appreciate your feedback and address your concerns below:
>
> >As the proposed algorithm is based EM, it is nice to have dicussions on its convergence conditions.
>
> Since our method is derived from an EM framework, the existing convergence guarantee of (generalized) EM methods also applies to our setting. Notably, it is known that under fairly general conditions on the monotonicity of the likelihood with respect to posterior $q$ in each iteration, the likelihood values in (generalized) EM converge to stationary points with zero gradients (see e.g., Section 3.1 of [1]). Therefore, our training scheme also shares a similar guarantee, i.e., converging to a stationary point of the empirical error of LEAF. We have added this discussion to Section 2.3 and included more technical details in Appendix A.4 in the revised paper.
>
> [1] McLachlan et al. The EM algorithm and extensions. 2007.
>
> >Hidden conflicts defined in this paper resembles graphical models with a latent variable, although graphical models cast stronger assumptions on the data generation process. Some dicussion on this topic is desired.
>
> Graphical models are a broad set of probabilistic models that use graphs to express the conditional dependence structure between variables of interest. In our work, we assume that conflicting data is generated through a general two-stage procedure with domain index as a latent variable with no additional structural assumptions. In principle, our formulation of data generation can indeed be viewed as a latent variable model that is tied to our problem setup, but several key assumptions on data and design choices differ our approach from common latent variable modeling approaches that are not designed for the conflicting data problem. Please refer to our response to Reviewer zrsD for more details.
>
> ---
> **Please let us know if our explanations above are still unclear, or if you have any other questions or concerns. Thank you again for your review!**

---

### Official Review · Reviewer_zrsD · 2022-10-26

**Confidence:** 4
**Correctness:** 4
**Technical Novelty And Significance:** 2
**Empirical Novelty And Significance:** 2
**Recommendation:** 6

**Clarity, Quality, Novelty And Reproducibility:**

See above.

A plus of the paper is that the writing is unusually clear. Though, I do think the focus of comparisons is not quite right to really clearly communicate the significance of the results.

**Strength And Weaknesses:**

For context, it may help to consider two closely related cases:
1. (latent variable modeling) We know that each data point is drawn according a hierarchical scheme that first draws a distribution, then draws the data point from this distribution. In this case, the obvious thing to do would be to set up learning via a latent variable model. This corresponds to the case where batch_size=1 in the present paper.
2. (unsupervised adaptation) We observe batches drawn from a number of domains, but we don't know a priori the relationship between domains. In particular, we don't know that there are only K distinct kinds of possible relationship. Then, the natural approach is to build a model that, given a new set of inputs, self-modifies to improve predictions on these inputs; see https://arxiv.org/abs/2102.12206 for an example of this kind of approach, and https://arxiv.org/abs/2112.05090 for a benchmark of such methods. In the present paper, this is done by guessing which of the K classifiers should be used for the test batch.

With this context, the main strength of the paper is that the particular setting it considers---data is observed as batches, and we know each batch is from a fixed domain---seems to add in extra structural assumptions in a manner that is realistic for some applications, and which may allow for better methods or analysis. The particular approach they take here (use EM to alternate assigning batches to classifiers and updating the classifiers according to batch assignment) is straightforward and reasonable.

However, in this view, the main weakness of the paper is that doesn't really focus on what's interesting about their data setup. Specifically, the natural baselines for comparison are 1 (latent variable modeling) and 2 (unsupervised domain adaptation). But it seems to me that the discussion, theory, and experiments all comparatively neglect these comparitor cases. That makes it both hard to assess the importance of the method, and hard to know what the important conceptual takeaways of the paper are. Generally, I think the paper needs to be substantially reworked to make all this clear.

As particular instances,
1. How does knowing that some pairs share the same domain help with identification (relative to straight latent variable modelling)?
2. How does the assumption that that there are only K domains help relative to general unsupervised domain adaptation? E.g., in terms of the PAC rate and in terms of empirical performance

Some further issues I see:
1. Theorems 1 and 2 seem trivial.

2. It's actually not obvious that ERM should fail in this data setup. The optimal classifier is P(Y| x) = \sum_k p(k | x) P(Y | k, x), where k denotes the underlying domain. If p(k|x) is learnable from the data, then a suitable ERM procedure should succeed. The claim here is presumably something like p(k|x) is not learnable given only single sample batches, but is learnable using multi sample batches. But, if that is the claim, I'm not able to find it spelled out clearly anywhere in the paper.

3. In the experiments, it's not clear to me what "ERM" means. In particular, is the capacity of the ERM model the same as the LEAF model? A fair comparison would be to run the EM algorithm, but eliminate the information about which datapoints come from the same batch---is that what you're doing? If so, it wasn't clear to me


**Summary Of The Paper:**

This paper considers supervised learning in the setting where the data is drawn in a 2 stage process: first, a domain is selected, second, a batch of data is drawn from this domain. The observations include the batch index, but not the underlying domain the data was collected from. The learning problem considered here is to find 1. a predictor that is optional for each domain, and 2. a rule for assigning a batch to a domain. The paper proposes a straightforward method for doing this.

**Summary Of The Review:**

Although I think the data setup is interesting and potentially fruitful, I don't think the results of the paper in its current form really communicate this. Minimally, the paper would require making it precisely clear how the batch size > 1 case helps vs simple latent variable modeling.

---

> ### Author Response · Authors · 2022-11-11
> **Response to Reviewer zrsD (4/4)**
>
>
> Addressing your detailed concerns:
>
> >Theorems 1 and 2 seem trivial.
>
> We agree that the results of Theorems 1 and 2 are natural by themselves. However, these results are necessary for the integrity of our analysis. For example, without the identifiability result in Theorem 1, bounding the generalization error would not justify our training objective since conflicting domains may not be identified even with zero generalization error. In practice, the results of Theorems 1 and 2 are also useful for designing heuristics to determine the number of low-level models when we have no prior knowledge of the number of conflicting domains (see the discussion in Appendix E.3 for more details).
>
> >It's actually not obvious that ERM should fail in this data setup. The optimal classifier is P(Y| x) = \sum_k p(k | x) P(Y | k, x), where k denotes the underlying domain. If p(k|x) is learnable from the data, then a suitable ERM procedure should succeed. The claim here is presumably something like p(k|x) is not learnable given only single sample batches, but is learnable using multi sample batches. But, if that is the claim, I'm not able to find it spelled out clearly anywhere in the paper.
>
> We do not claim that $p(k|x)$ is not learnable given single _unlabeled_ examples but learnable given batches of _unlabeled_ examples. Instead, in our problem setup $p(k|x)$ is never learnable without having access to $y$ -- as we have mentioned in our above response, the crux of the conflict between domains is that **different domains label the same set of $x$ differently (as in Definition 1), hence the marginal distribution $p(x)$ without $y$ cannot be used to decide whether domains are conflicting**. This is why the _allocation function_ $g$ in our LEAF objective has access to _both $x$ and $y$_.
>
> >In the experiments, it's not clear to me what "ERM" means. In particular, is the capacity of the ERM model the same as the LEAF model? A fair comparison would be to run the EM algorithm, but eliminate the information about which datapoints come from the same batch---is that what you're doing? If so, it wasn't clear to me.
>
> - The ERM baseline in our experiments (both regression and classification) corresponds to training a _single_ model on _all_ training data (i.e., without considering the conflict between domains) using standard squared or cross-entropy loss. This definition is consistent throughout our paper and directly follows the standard definition of ERM [1].
>
> - In our experiments, we have also compared LEAF with the variant that uses standard EM (denoted by LEAF-EM) and observed that LEAF outperformed LEAF-EM often by a large margin. However, both LEAF and LEAF-EM used batches of inputs. Therefore, as requested, we also added a new baseline (denoted by **EM($n=1$)**) that (1) uses standard EM and (2) eliminates the information about which datapoints come from the same batch. We evaluated its performance on the CIFAR100 dataset (with two low-level models), following the same setup as in the main text. However, _this baseline cannot converge to a stable solution, and we found its performance has large fluctuations across different runs and is generally much worse than both LEAF and LEAF-EM_. This validates the example on the identifiability issue in the $n=1$ case in our above response. Detailed results are in the following table:
>
> |Method|Superclass error (%)|Class error (%)|
> |:--:|:--:|:--:|
> |LEAF|21.40|25.05|
> |LEAF-EM|32.71|33.83|
> |EM ($n=1$), 1st run|61.71|52.46|
> |EM ($n=1$), 2nd run|67.43|55.20|
> |EM ($n=1$), 3rd run|71.33|48.79|
>
> For each domain (superclass/class), we report the **error rate (%)** of the low-level model that performs _best_ on this domain. We can observe that the result of EM ($n=1$) is unstable across different runs, and both LEAF and LEAF-EM with $n>1$ perform much better.
>
>
> [1] The nature of statistical learning theory. Vapnik. 1999.
>
> ---
> **Please let us know if our explanations above are still unclear, or if you have any other questions or concerns. Thank you again for your review!**

---

> ### Author Response · Authors · 2022-11-11
> **Response to Reviewer zrsD (3/4)**
>
> **Regarding the second major concern**, we note that _we have already discussed the difference between our problem setup and unsupervised domain adaptation in **Appendix F**_. In response to your concern, we have expanded the corresponding paragraph in the revised paper to provide a more detailed discussion. **In general, our problem setting and goal are essentially _orthogonal_ to those of unsupervised domain adaptation, and unsupervised domain adaptation methods _provably_ cannot handle conflicting data in our setup**:
>
> - The formulation of unsupervised domain adaptation focuses on the (probably test-time) **adaptation** process of the model trained on one or multiple source domain(s) to one or multiple target domain(s). In other words, **unsupervised domain adaptation does not consider _training_ on the sources domains _itself_ to be an issue**, which is not the case in our setup if source domains contain conflicting data -- we cannot train a single model that is simultaneously optimal for all source domains when they conflict. By contrast, our problem setup focuses exactly on the _**training**_ process in the presence of conflicting data, rather than the adaptation process as unsupervised domain adaptation does. Therefore, **the assumption of the existence of $K$ domains during training is not for helping general unsupervised _adaptation_, but is for formulating the _training_ process when conflicting data exist**. In fact, LEAF does **not** perform test-time adaptation as the approach [1] mentioned in your review; our evaluation protocols (see **Sections 4.1 and 4.2** for details) are also designed to examine whether the learner successfully learns all target functions in conflicting domains, rather than evaluate the adaptation ability of the learner to a new domain.
>
> - **The data setup of our work and unsupervised domain adaptation is also fundamentally different**: a common assumption adopted by most of unsupervised domain adaptation research is the **covariate shift** assumption, i.e., _the input-conditioned label distribution $p(y|x)$ is the **same** for all domains while the input marginal distribution $p(x)$ varies_. The WILDS benchmark [2] mentioned in your review gives multiple instances of this assumption, e.g., the iWildCam dataset defines the task of classifying animal species based on camera trap photos and defines domains as different cameras, where the photo-specie relation stays the same across different domains. By contrast, we define conflicting domains as that **the target functions from different domains give distinct outputs for the _same_ set of inputs, which represents _varying_ $p(y|x)$**. For example, the _same_ photo of a shark can be labeled as "shark" in one domain while being labeled as "fish" in another domain.  Therefore, the setup of unsupervised domain adaptation and our work are orthogonal. We also note that from a theoretical view, existing domain adaptation methods are **provably infeasible** in our problem since their success _requires the existence of a model with a low **joint** error on all domains_ (see e.g., [3,4]), which does not hold if $p(y|x)$ varies a lot across different domains.
>
> - Another minor difference is that in unsupervised domain adaptation, the domain label of each example is usually given during training, while our work weakens this assumption by only assuming that the examples in each batch are drawn from the same domain.
>
> [1] PADA: Example-based prompt learning for on-the-fly adaptation to unseen domains. Ben-David et al. 2021. [https://arxiv.org/pdf/2102.12206.pdf](https://arxiv.org/pdf/2102.12206.pdf)
>
> [2] Extending the WILDS benchmark for unsupervised adaptation. Sagawa et al. 2021. [https://arxiv.org/pdf/2112.05090.pdf](https://arxiv.org/pdf/2112.05090.pdf)
>
> [3] Impossibility theorems for domain adaptation. Ben-David et al. 2010. [https://proceedings.mlr.press/v9/david10a/david10a.pdf](https://proceedings.mlr.press/v9/david10a/david10a.pdf)
>
> [4] Domain adaptation: Learning bounds and algorithms. Mansour et al. 2009. [https://arxiv.org/pdf/0902.3430.pdf](https://arxiv.org/pdf/0902.3430.pdf)

---

> ### Author Response · Authors · 2022-11-11
> **Response to Reviewer zrsD (2/4)**
>
>  **(2)** Consider a classification problem that follows the CIFAR100 setup in our experiments, where each $x$ corresponds to the input image and $y$ corresponds to the label. There are two domains, $d_1$ and $d_2$, where images in $d_1$ are labeled by superclass names, while images in $d_2$ are labeled by class names. As an illustrative example, we consider $d_1$ with labels "fish" and "flowers" and $d_2$ with labels "shark", "flatfish", "roses" and "sunflowers", with the first two labels belonging to "fish" and the last two labels belonging to "flowers". Both $d_1$ and $d_2$ contain input images that belong to "shark", "flatfish", "roses" and "sunflowers" classes with the same input distribution. Then, in the $n=1$ setting, both of the following two solutions achieve zero training error. **S1:** model $h_1$ classifies all images by their superclass labels, while model $h_2$ classifies all images by their class labels. **S2:** model $h_1$ classifies fish images according to their superclass labels and classifies flower images by their class labels, while model $h_2$ classifies fish images by their class labels and classifies flower images by their superclass labels. In other words, S1 recovers the target function in both domains, while S2 fails to recover any target function since the outputs of both models $h_1,h_2$ in S2 are _mixtures_ of superclass labels and class labels. Similar to the regression case, $n>1$ can prevent S2 from being an optimal training solution since superclass labels "fish" and "flowers" can be sampled in the same batch and thus should be predicted by the same model. _In the revision, we have added experiments on the CIFAR100 dataset using $n=1$ and validate our above construction_; please find these experimental results in **Appendix G** as well as our response to your detailed concerns below.
>
> Besides the concrete failure cases we provide above, **our theoretical result on the generalization error bound of LEAF also sheds light on the failure of $n=1$:** to bound the generalization error between empirical and expected allocation functions (Eq. (8c) in Theorem 3), we generally require a sufficiently large $n$. Therefore, $n=1$ may create a non-negligible gap between empirical (training) error and expected error. Since our Theorem 1 suggests an equivalence between domain identification and a small _expected error_, having a gap between empirical and expected errors can pose difficulty in domain identification.
>
>  - **Necessity of imposing the unimodal constraint on the E-step output:** our method differs from standard EM in that we enforce a _unimodal_ E-step by assuming the model posterior as a Kronecker-Delta distribution over the model set. While this constraint itself is not new, this design choice is tied to our problem setup and thus differs LEAF from "straight latent variable modeling" with no problem-specific design. We empirically demonstrate that this design choice brings significant performance gains over standard EM (denoted by "LEAF-EM" in our experiments); please check our result tables (Tables 1, 2 in Section 4 and Table 3 in Appendix E.1) for these results.

---

> ### Author Response · Authors · 2022-11-11
> **Response to Reviewer zrsD (1/4)**
>
>
> Thank you for your detailed feedback! Your review expressed two major concerns:
> 1. Comparison to latent variable modeling; in particular, what are the benefits of observing a batch of data with batch size $n>1$ compared to the case of $n=1$.
> 2. Comparison to unsupervised domain adaptation methods.
>
> In the following, we will address these two major concerns first and then respond to the other points. **While we very much appreciate your comments and find them helpful for improving and contextualizing our work, some of the technical concerns you have raised suggest that you may have misunderstood certain parts of our setup and results.** In this rebuttal we will try to carefully address them, and we hope that you can consider increasing the score if we successfully address your major concerns.
>
> **Regarding the first major concern**, we very much agree that discussing the relation between our work and latent variable modeling is helpful for better locating our work; in response, _we have added a separate paragraph on this point (**Appendix F**), and a separate section (**Appendix G**) highlighting the importance of using a batch of data ($n>1$) rather than single data points ($n=1$)_. We have added explicit references to these sections in the main text in both Section 2.1 and Section 2.3. Here we summarize our discussion on these points:
>
> In a broad sense, our method falls into the category of latent variable modeling by treating the domain identity of each data pair as a (discrete) latent variable. However, the concept of "latent variable modeling" itself is rather general, and distinct methods may be developed based on the specific problem setup. Here we would like to underscore the following two points:
>
> - To the best of our knowledge, **our problem setup on learning from conflicting data is novel and has not been formulated or addressed by prior methods with/without latent variable modeling**.
>
> - **Our method is tailored to our problem setup**, and several key assumptions and design choices are crucial for solving our problem:
>
>   - **Necessity of using batches of data ($n>1$) rather than single data pairs ($n=1$):** In general, $n=1$ poses fundamental difficulty in identifying conflicting domains; **in the worst case, $n=1$ would make identification of conflicting domains impossible**. Here we construct two examples to illustrate this impossibility, one for regression and one for classification, both of which directly follow our experimental setup:
>
>     **(1)** Consider a regression problem with two domains, namely $d_1$ and $d_2$, where training $(x,y)$ pairs are uniformly sampled from $y_1=x,x\\in[0,1]$ in $d_1$ and $y_2=1-x,x\\in[0,1]$ in $d_2$, respectively. Then, in the $n=1$ setting, both of the following two solutions achieve zero training error.
>     **S1:** $h_1(x)=x,x\in[0,1]$ and $h_2(x)=1-x,x\in[0,1]$;
>     **S2:** $h_1(x)=\left\\{\begin{array} & x, &x\\in[0,0.5)\\\\ 1-x,&x\in[0.5,1]\end{array}\right.$ and $h_2(x)=\left\\{\begin{array} &1-x,& x\in [0,0.5)\\\\ x,&x\in[0.5,1] \end{array}\right.$. However, only S1 is the desired solution that separately recovers $y_1$ by $h_1(x)$ and recovers $y_2$ by $h_2(x)$, while S2 does not recover either $y_1$ or $y_2$ by any of the models. When $n=1$, S1 and S2 are not identifiable since both of them are optimal with zero training error. By contrast, $n>1$ would effectively prevent the undesired solution S2, since we may sample two points $(x^1,y^1)$ and $(x^2,y^2)$ from the same domain with $x^1\in[0,0.5)$ and $x^2\in[0.5, 1]$ in the _same_ batch so that S2 does not achieve zero training error anymore (since all examples in the same batch are assigned to the _same_ model). _We did observe such cases in our regression experiments (see **Figure 6(c)** in **Appendix E.3**)_, where $n=1$ led to incorrect identification of domains.

---

> ### Author Response · Authors · 2022-11-17
> **Looking forward to your feedback!**
>
> Dear Reviewer zrsD,
>
> Thank you again for your detailed review and constructive comments. Since the official discussion period is ending soon, we would like to kindly remind you to check our response and the revised paper. Any feedback would be appreciated. Thank you!

---

> > ### Comment · Reviewer_zrsD · 2022-11-28
> > **increasing score, but not fully convinced**
> >
> > The rebuttal seems to miss the forest for the trees.
> >
> > The crux of my issue with the paper is simply:
> > 1. the novelty of the paper is the data setup --- each element of a batch shares an environment, but it's not known what the environment is for each batch
> > 2. then, the results of the paper should be aimed at elucidating exactly what's interesting about this data setup---what it buys and what it doesn't, but
> > 3. the results don't really aim at this. Instead, they just establish properties of LEAF de novo.
> >
> > Importantly, my issue is with the presentation and emphasis of the paper, not its novelty per se. I understand already that the setup here is different than traditional ones, I'm saying that the task for the paper is to clearly explain how it's different and why that's important. This is the point on which I feel it falls short.
> >
> > However,
> >
> > I did previously misunderstand the evaluation procedure. I now understand that you're looking at batches of data from entire domains, and then choose the LEAF model that does best here. In this case, your baselines seem more reasonable to me --- though I think looking example by example and computing regret would be fairer and more standard than using an entire batch to select the best of the available models.
> >
> > I also previously did not appreciate theorem 3, and term 8c, which I think comes close to saying something about the importance of the domain-unknown setting.
> >
> > Accordingly, I'm increasing my score to a weak accept. I still think this paper is significantly weaker than it would be if it focused more narrowly on elucidating the interesting aspects the data setup, but it does seem good enough to publish in its current state.

---

> > > ### Author Response · Authors · 2022-11-30
> > > **Thank you for the feedback; author response**
> > >
> > > Thank you very much for your feedback! **We are very glad to hear that we have successfully resolved your misunderstanding on our experiment part**. _Partially due to that we had not located the exact cause of the main confusion, we responded to your concerns in a point-by-point manner_, without adding much discussion on the big picture. In light of your follow-up comments, below we reiterate some key aspects of our data setup to hopefully clarify that **our setup differs from existing ones in a way that is both interesting and practically meaningful**. We will also rephrase corresponding paragraphs in the paper accordingly.
> > >
> > > - We believe that summarizing the novelty of our data setup as "each element of a batch shares an environment, but it's not known what the environment is for each batch" is somewhat **incomplete**. This indeed describes our main assumption on data sampling, but does not include our key assumption on data -- that the data from different environments (domains) are potentially **conflicting**. _This point is very important as it leads to essential distinctions between our setup and existing setups (e.g., unsupervised domain adaptation, UDA) and elicits several interesting characteristics about our setup (e.g., the distinction between batch size $n=1$ and $n>1$ with respect to domain identifiability)._ Meanwhile, this assumption of conflicting data is motivated by practical scenarios (see Section 1 for more details on the motivation and some exemplar cases in Section 4.2) where methods based on different data setups such as UDA do not apply, while LEAF may come in handy.
> > >
> > > - On the other hand, our experiments are **intentionally designed** to not only show the properties of LEAF (i.e., validating our theoretical claims), _but also show LEAF's applicability in different practical settings of interest and shed light on how we can transfer our data setup to practical prediction problems_ (also see our response to reviewer 4gxX) -- to make this clear, we discussed some exemplar cases in Section 4.2. As mentioned by other reviewers, we also benchmarked LEAF against problem-specific, competitive baselines and demonstrated LEAF's advantage, so we feel that it may be a bit unfair to say that our experiments "just establish properties of LEAF de novo".
> > >
> > > Finally, we would like to emphasize that **we do really appreciate your efforts in the review, which led to some valuable discussions between the authors and have definitely helped us to improve our work both conceptually and empirically.**

---

### Decision · Program_Chairs · 2023-01-20

**Decision:**

Reject

**Justification For Why Not Higher Score:**

the novelty of the paper that motivated relatively high scores by the reviewers was questioned by a more detailed analysis of prior work and a consensus was reached to reject the submission.

**Justification For Why Not Lower Score:**

N/A

**Metareview: Summary, Strengths And Weaknesses:**

After the authors' responses, there was an agreement between reviewers that the paper presents a valuable contribution. Even though the reviewers pointed out some weaknesses of the paper, overall they agreed that the main strength of the paper is the novelty or practical value of the framework.

After further discussion however, more attention was given to the prior work by Su et al. "Task Understanding from Confusing Multi-task Data" (ICML 2020). Looking more closely at this prior work, it seemed that:

1- The setup, solutions and experiments are very similar. Both papers use a task assignment process that is performed during the training of task-specific classifiers with an EM-style approach. It seems that the main differences are that the current submission works with batches and performs per-batch task assignment while the previous paper works on the basis on individual samples and learns a network to predict a soft task assignment. Even though there are new tasks in the experimental evaluation, the datasets are the same and tasks are similar.

2- The way the Su et al is cited in the submission mislead the reviewers to believe that this prior work was not closely connected to the present submission. However, there is a consensus among reviewers that Su et al. is very close to the current work.

In  the end, during the disucussion phase the reviewers revised their opinion regarding the novelty of the paper, which is now assessed as weak by all of them. Also, before any publication, it seems necessary to thoroughly discuss Su et al.'s paper in the future iterations of this paper and better explain the novelty, differences and the practical relevance of these differences.